# Allopolyploid origin and diversification of the Hawaiian endemic mints

Crystal M. Tomlin[1], Sitaram Rajaraman [2,3], Jeanne Theresa Sebesta[1], Anne-Cathrine Scheen[4], Mika Bendiksby[5], Yee Wen Low [6], Jarkko Salojärvi[2,3], Todd P. Michael [7], Victor A. Albert [1] ✉ & Charlotte Lindqvist [1] ✉

Island systems provide important contexts for studying processes underlying lineage migration, species diversification, and organismal extinction. The Hawaiian endemic mints (Lamiaceae family) are the second largest plant radiation on the isolated Hawaiian Islands. We generated a chromosome-scale reference genome for one Hawaiian species, *Stenogyne calaminthoides*, and resequenced 45 relatives, representing 34 species, to uncover the continental origins of this group and their subsequent diversification. We further resequenced 109 individuals of two *Stenogyne* species, and their purported hybrids, found high on the Mauna Kea volcano on the island of Hawai'i. The three distinct Hawaiian genera, *Haplostachys*, *Phyllostegia*, and *Stenogyne*, are nested inside a fourth genus, *Stachys*. We uncovered four independent polyploidy events within *Stachys*, including one allopolyploidy event underlying the Hawaiian mints and their direct western North American ancestors. While the Hawaiian taxa may have principally diversified by parapatry and drift in small and fragmented populations, localized admixture may have played an important role early in lineage diversification. Our genomic analyses provide a view into how organisms may have radiated on isolated island chains, settings that provided one of the principal natural laboratories for Darwin's thinking about the evolutionary process.

Organismal radiations are unique cases in which species exhibit great morphological and ecological diversity despite limited genetic differentiation. Many such diversifications have been termed adaptive radiations[1], even in the absence of clarity on whether adaptive forces primarily led to species differentiation. The Hawaiian Islands are an exemplar natural laboratory to study radiations and investigate their possible adaptive nature, due to their habitat diversity and isolated location over 3700 km from any substantial landmass. The island chain comprises a series of volcanoes that formed sequentially via movement of the Pacific Plate over a mantle hot spot, and they have well-

characterized formation and erosional age profiles, both among them and within given islands[2,3]. On the youngest and largest island, Hawai'i, the creation of new habitats and dissection of older ones by lava flows is observable in real time. This continuous eruption-based fragmentation over geological time of both pioneer and old-growth habitats, and the age and ecological/geological gradients within the high islands themselves, has no doubt promoted speciation in many lineages. Indeed, the Hawaiian Islands feature one of the most dramatic rainfall gradients in the world[4], providing incredibly diverse environments from alpine deserts to wet forests[5]. The question, however, remains:

[1]Department of Biological Sciences, University at Buffalo, New York, USA. [2]School of Biological Sciences, Nanyang Technological University, Singapore, Singapore. [3]Organismal and Evolutionary Biology Research Programme, Faculty of Biological and Environmental Sciences, University of Helsinki, Helsinki, Finland. [4]Stavanger Botanic Garden, City of Stavanger, Norway. [5]Natural History Museum, University of Oslo, Oslo, Norway. [6]Singapore Botanic Gardens, National Parks Board, Singapore, Singapore. [7]The Plant Molecular and Cellular Biology Laboratory, Salk Institute for Biological Studies, La Jolla California, USA. ✉e-mail: vaalbert@buffalo.edu; cl243@buffalo.edu

are bursts of speciation, e.g., in the Hawaiian Islands, due principally to geographic (neutral) speciation, to ecological (adaptive) speciation, or a combination of both? There is also the question of what genomic features might underpin a successful organismal radiation. Two commonly proposed phenomena are hybridization[1] and polyploidization[6,7], which can of course occur together. In the case of hybridization, incomplete lineage sorting (ILS) of genetic variation that defies the species phylogeny can leave a similar signature of disrupted monophyly, necessitating careful distinction between the two phenomena. This is especially important in the case of rapidly radiating lineages, in which time for ancestral polymorphisms to completely sort by lineages is often extremely short[8].

Despite great interest in the evolution of organismal radiations, only one tenth of (presumed) adaptive radiation studies focus on plants[9]. Of these relatively few plant studies, the vast majority have used limited DNA markers, or reduced genomic representations for phylogenomic reconstruction[10], which frequently do not provide the necessary resolution to understand evolutionary histories complicated by features such as whole genome duplication, paralogy, hybridization, and rapid genetic divergence[11]. With such rapid divergence and little genetic differentiation it can be challenging to distinguish ILS from hybridization using either maternally inherited (even whole plastome) or single-copy nuclear loci. Hawaiian plant radiations studied to-date, including the silverswords[12], lobeliads[13,14], and endemic mints[15–19], have mostly fallen into this experimental realm, although recent research, including the o'hia tree, *Metrosideros polymorpha*[20,21], and the species-rich Southeast Asian *Syzygium*[22], has taken advantage of a reference genome and large-scale genome resequencing.

Among the most species-rich angiosperm radiations to occur on the Hawaiian Islands is that of the Hawaiian endemic mint lineage (Lamiaceae), which exhibit considerable diversity in fruit, floral, and habitat features leading to their classification into ~60 species assigned to the three genera, *Haplostachys*, *Phyllostegia*, and *Stenogyne*[23,24]. *Haplostachys*, which consists of five species with only a single extant taxon, *H. haplostachya*, is the only genus with dry fruits; the other genera bear fleshy fruits, which are rare in Lamiaceae. *Haplostachys* and *Phyllostegia* (34 species) both have mostly white to pink, fragrant flowers with a prominent lower corolla lip, typically associated with insect pollination. *Stenogyne* (21 species) is unique in that it has primarily yellow to pink-purple nectar-producing flowers with a reduced lower lip and a longer corolla tube, typical of bird pollination. The distribution of *Haplostachys* has today been reduced to relatively small, restricted subpopulations in xerophytic shrubland at low-mid elevation between Mauna Loa and Mauna Kea on Hawai'i[25]. *Phyllostegia* and *Stenogyne* are widely distributed on all the extant high islands of Hawaii, primarily occurring in wet/mesic forest environments, although a few species of *Stenogyne* are found in subalpine zones of Haleakala, Mauna Kea, and Mauna Loa[24]. Most individual species are confined to just a single island in the archipelago.

Prior phylogenetic analyses of the Hawaiian mint lineage, based on single or few loci, discovered that the Hawaiian endemic mints form a monophyletic group nested within the nearly global genus *Stachys*[15]. *Stachys* is the most species-rich genus within the tribe Stachydeae, a tribe within Lamiaceae subfamily Lamioideae that exhibits evolutionary complexities such as a broad range of ploidy levels, frequent natural hybridization events, and paraphyletic genera (of which *Stachys* itself is a prominent example)[26,27]. According to allelic data and chromosome counts, the Hawaiian mints are likely paleo-octoploids, bearing evidence for two polyploidy events[17] following the ancient *gamma* hexaploidy event at the base of all core eudicots[28]. Further studies indicated that the initial colonizer of the Hawaiian Islands may have been of allopolyploid origin, as suggested by phylogenetic discordance between genetic markers, where plastid sequences showed the lineage to be most closely related to meso-South American *Stachys*, while nuclear markers linked the clade to temperate North American

*Stachys* species[15,18,19,29]. In addition to a hybrid origin for the lineage, there have been reports of ongoing interspecies admixture at Mauna Kea on the Big Island, Hawai'i[24]. Hence, the Hawaiian mints are a remarkable case not only of morphological radiation, but also polyploidization, past and present hybridization, and as a model system in which to study the role of these phenomena in evolutionary diversification on oceanic islands.

In this work, our three major goals include: (i) generate a high-quality, chromosome-level genome of *Stenogyne calaminthoides* (Fig. 1a) to investigate polyploid history of the Hawaiian mint lineage, (ii) use this reference genome and resequencing of multiple species to establish the origin and phylogeny of the Hawaiian mints and their mainland relatives, as well as potential admixture history, and (iii) investigate recent introgression among *Stenogyne* species that co-occur at high elevation on Mauna Kea[24]. Using these genomic resources, we evaluate whether geographic speciation, a drift-biased process that may limit adaptive interpretations, played a major role in the Hawaiian mint radiation. We show that an allopolyploidy event is shared among the Hawaiian mints and their closest North American ancestors, and that the Hawaiian lineage principally diversified by parapatry and drift in small and fragmented populations, while localized admixture may have played an important role early in lineage diversification.

## Results and discussion

### Reference genome characterization unveils a repeat-dense genome

The genome size of *Stenogyne calaminthoides* was first predicted to be ~1.2–1.6 billion bases (Gb) based on $k$-mer analyses of Illumina sequences. The initial assembly of *Stenogyne calaminthoides*, using over 75 Gb of Nanopore reads greater than 35 thousand bases (kb) in length, resulted in an assembly of ~2.4 Gb. This genome size of approximate diploid value suggested that our primary assembly may have contained two divergent haplotypes. Visualizing the assembly graph, we identified a large "hairball" of complex genomic features and repetitive sequences[30], perhaps corresponding to a relatively recent transposable element (TE) burst (Supplementary Fig. 1). This initial genome assembly consisted of over 77% repetitive elements, 46% of which correspond to long terminal repeat (LTR) retrotransposons (Supplementary Data 1). After filtering diploid regions from the assembly, the resulting haploid assembly reduced to ~1.4 Gb in size, consistent with the $k$-mer-based haploid estimates. After scaffolding with Hi-C reads (Fig. 1b), the final assembly had an N50 over 37 Mb and contained 434 scaffolds, including 32 scaffolds larger than 25 Mb, closely matching the expected chromosome count of $n = 32$[31] (Fig. 1c, Supplementary Data 2). The unannotated assembly had a BUSCO (Benchmarking Universal Single-Copy Orthologs) completeness score of 94.9% (49.4% complete single copy and 45.5% duplicated genes) (Supplementary Data 3). Annotation of the genome resulted in a total of 77,090 genes with a BUSCO score of 86% (consisting of 35.3% single copy and 50.7% duplicated genes) and an N50 of 4,162 bp (Supplementary Data 3). The large number of genes (and high percentage of BUSCO duplicates) may reflect an incompletely fractionated (diploidized) gene space following a recent whole genome duplication (WGD) event (discussed further below)[32], as has been noted for polyploid species, including the paleo-octoploid Lamiaceae species *Pogostemon cablin*, which had a reported 110,850 genes total[33].

### Genome architecture and polyploid history within Lamiales is marked by a shared ancient allopolyploid event

The *Stenogyne calaminthoides* genome was compared with several other representative core eudicot genomes using syntenic depth FractBias plots[34]. Such plots are a useful means to query not only ploidy depths, but also potential subgenomic dominance biases at the time a WGD occurred[35]. Fractionation bias patterns of *Stenogyne*

*calaminthoides* mapped against grapevine (*Vitis vinifera*) showed 4:1 synteny for the two species, respectively (Fig. 1d). This is strong evidence, along with the accompanying syntenic dotplot (SynMap) visualization (Supplementary Fig. 2), that the Hawaiian mint species underwent two sequential WGD events since common ancestry with *Vitis*. Of the four *Stenogyne* genomic blocks mapping to grapevine chromosomes, there are two pairs of blocks wherein each member of a pair follows the other closely in gene retention (fractionation) percentage, i.e., presenting little evidence for diploidization bias. Each of these pairs are in turn broadly separated in fractionation percentage, suggesting substantial subgenome dominance during the earlier WGD event. Distinct epigenetic masking in progenitor genomes of allopolyploid lineages can lead to considerable purifying selection differences between subgenomes, resulting in dominant/recessive (biased) patterns. Unbiased fractionation between subgenomes can in turn reflect little masking difference, or even autopolyploidy[36,37]. As such, the most recent, less biased WGD visible in *Stenogyne* may have involved a narrow cross, whereas the earlier polyploidy event likely involved divergent progenitor species.

Other sequenced Lamiales genomes also show signatures of ancient WGD since ancestry with grapevine. The 1KP Project (the One Thousand Plant Transcriptomics Project or OTPTP) reported a WGD event (denoted as ANMAα) shared by many core Lamiales families including Lamiaceae[38]. In contrast, other investigators reported that the WGD events of the Lamiaceae species *Callicarpa americana* and *Tectona grandis* were independent based on synonymous substitution (*K*s) distributions for paralogous gene pairs, and that *Tectona* experienced an additional ploidy event[39]. However, this determination based on *K*s alone may have been flawed without macrosyntenic analysis, and stood in contrast to results from the *Tectona* genome[40]. Our comparisons of the *Stenogyne* genome with other Lamiales species support the findings of the 1KP Project, suggesting that the early WGD in *Stenogyne* was likely a single event shared with *Callicarpa* and *Tectona*, as well as many other core Lamiales, such as *Mimulus guttatus* (Phrymaceae) (Fig. 1e). For example, the fractionation patterns of *Callicarpa* and *Tectona* mapped to *Vitis* are remarkably similar, as these two Lamiaceae taxa both map 2:1 to *Vitis* and even share some chromosomal fusion events since common ancestry with that taxon (Supplementary Fig. 3a). Furthermore, fractionation bias between the two subgenomes of *Buddleja* (Scrophulariaceae) and *Mimulus* (the latter far outside Lamiaceae) are broad (Supplementary Fig. 3b), similar to the first WGD event in *Stenogyne*. Hence, we conclude the presence of one shared allopolyploidy event at the base of Lamiaceae and most other core Lamiales. Further, a second WGD in *Stenogyne calaminthoides* is apparent (syntenic depth plots showed up to 8X depth, while *Callicarpa* and *Tectona* had 6X depth; Supplementary Fig. 4), verifying the additional, more recent polyploidy event in *Stenogyne* since its split with these other members of Lamiaceae. Finally, syntenic dotplots of *Stenogyne* vs. *Callicarpa* and *Stenogyne* vs. *Tectona* (Supplementary Fig. 5) similarly showed 4:2 ratios, confirming a doubling of the *Stenogyne* genome since its split with these two relatives.

## Phylogenomic analyses reveal that Hawaiian mints are most closely related to western North American *Stachys*

To investigate relationships among the Hawaiian mints and their closest relatives in Stachydeae, we mapped Illumina reads for 45 mint

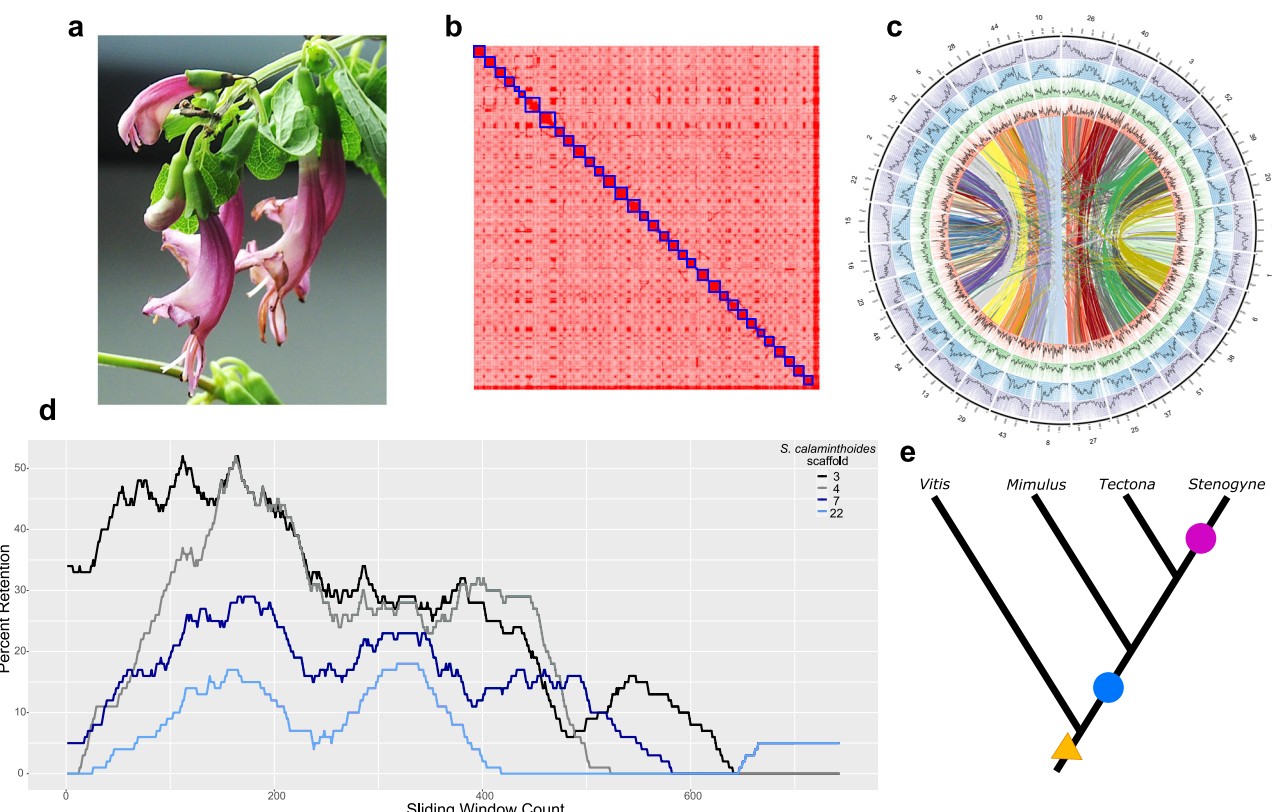

**Fig. 1 | Assembly and structural evolution of the *Stenogyne calaminthoides* genome. a** Photo of *Stenogyne calaminthoides* (credit: V.A.A.). **b** Hi-C contact map with inferred chromosomes in blue boxes. **c** Gene and repeat landscape of *S. calaminthoides* genome with outer to inner tracks showing genes (purple), *Copia* (blue), Helitron (green), and *Gypsy* (red) repeats, respectively. Density over 1 Mb intervals is shown by black lines. Regions of synteny are shown in the center and each pair shares a color. Scaffold number is shown on the exterior. **d** Fractionation bias plot of *S. calaminthoides* mapped onto *Vitis vinifera* chromosome 10. Each coloured line represents a *S. calaminthoides* scaffold. **e** Schematic phylogenetic tree showing inferred polyploid events in *Stenogyne*, with the triangle representing the gamma paleohexaploidy found in all core eudicots, the blue circle representing a whole genome duplication (WGD) found in most core Lamiales, and the pink circle representing the WGD event exclusive to *Stenogyne*. Source data for Figs. 1c and 1d are provided as a Source Data file.

samples to the *Stenogyne calaminthoides* reference genome and generated maximum likelihood phylogenies based on both nuclear genomic single nucleotide polymorphisms (SNPs) and assembled chloroplast genomes (Supplementary Data 4–7). Consistent with previous analyses[29], the chloroplast genome phylogeny demonstrated weak resolution of relationships among Hawaiian mints, as well as very short internal branch lengths, as expected of a rapid radiation (Supplementary Fig. 6). However, two main clades were strongly supported, one grouping Hawaiian mints with *Stachys coccinea* and one individual of *Stachys chamissonis*, and another clade containing eastern North American and Asian (ENAA) *Stachys*, including a second accession of *Stachys chamissonis*. Confirming previous findings[15,18,19], phylogenies based on nuclear data (Fig. 2, Supplementary Fig. 7) were discordant with the plastid tree in that (*i*) the closest relatives of Hawaiian mints were their western North American (WNA) *Stachys* relatives (*Stachys bullata*, two representatives of *S. chamissonis*, *S. ajugioides*, and two representatives of *S. albens*), (*ii*) the two *Stachys chamissonis* individuals appeared in the same clade, and (*iii*) *Stachys coccinea* was instead a distant relative. These relationships could be explained if a *Stachys coccinea*-like ancestor was the maternal progenitor of an allopolyploid lineage that colonized the Hawaiian Islands, and that the chloroplast genome was captured by *Stachys chamissonis* (at least the population bearing this plastome) in this hybridization event. The recentmost WGD visible in *Stenogyne*, if an allopolyploidy event, may have co-occurred with this plastome capture. However, the fact that all Hawaiian mints and a single WNA *Stachys chamissonis* individual share a plastid haplotype suggests a more complex scenario. It is possible that hybridization and polyploidization co-occurred multiple times independently, or that hybridization among allopolyploids followed WGD events, as has been suggested for *Dactylorhiza*[41]. This scenario may have been further confounded by subsequent ILS of the two plastid haplotypes.

In the SNP phylogeny, monophyly of the entire Hawaiian lineage was strongly supported (Fig. 2, Supplementary Fig. 7). Within the Hawaiian mints, two main clades of *Phyllostegia* were joined by a very short internal branch. Two *Phyllostegia warshaueri* samples and *P. racemosa* each had long external branches corresponding to unique variation of unknown origin, unsampled species diversity, or possibly extensive extinction along their lineages[42]. There was no strong pattern of phylogeographic structuring within *Phyllostegia* (Supplementary Fig. 8); however, the two species from Kaua'i (*P. electra* and *P. waimeae*) grouped together. Within *Stenogyne*, one first-branching clade consisted of samples found only on Kaua'i (these species are hereafter referred to as the Kaua'i clade). Sister to this clade, *Stenogyne* split into two additional clades, here referred to as the *Stenogyne rugosa* clade (also including *Stenogyne microphylla* and *S. bifida*) and the *Stenogyne macrantha* clade (also containing *Stenogyne calaminthoides*, *S. cranwelliae*, and *S. scrophularioides*). Interestingly, within the *Stenogyne macrantha* clade, not all samples of *S. macrantha* grouped together, possibly reflecting representation of two extreme morphological forms, as previously reported[24].

## Genes from de novo assembled genomes show *Phyllostegia* to be paraphyletic

Because SNP trees reflect a phylogenetic summary across the entire genome, we next used an independent method to generate a locus-by-locus coalescent species tree, both to avoid potential reference genome mapping bias[43] and to supplement the SNP-based results. We generated de novo Illumina-based genome assemblies from the 45 resequenced samples and extracted their conserved BUSCO genes (Supplementary Fig. 9). The average BUSCO completeness score was ~87% (Supplementary Data 8). The multilocus coalescent phylogeny, based on 1,336 BUSCO genes, agreed with the SNP tree in that WNA *Stachys* were the closest relatives to the Hawaiian mints, but

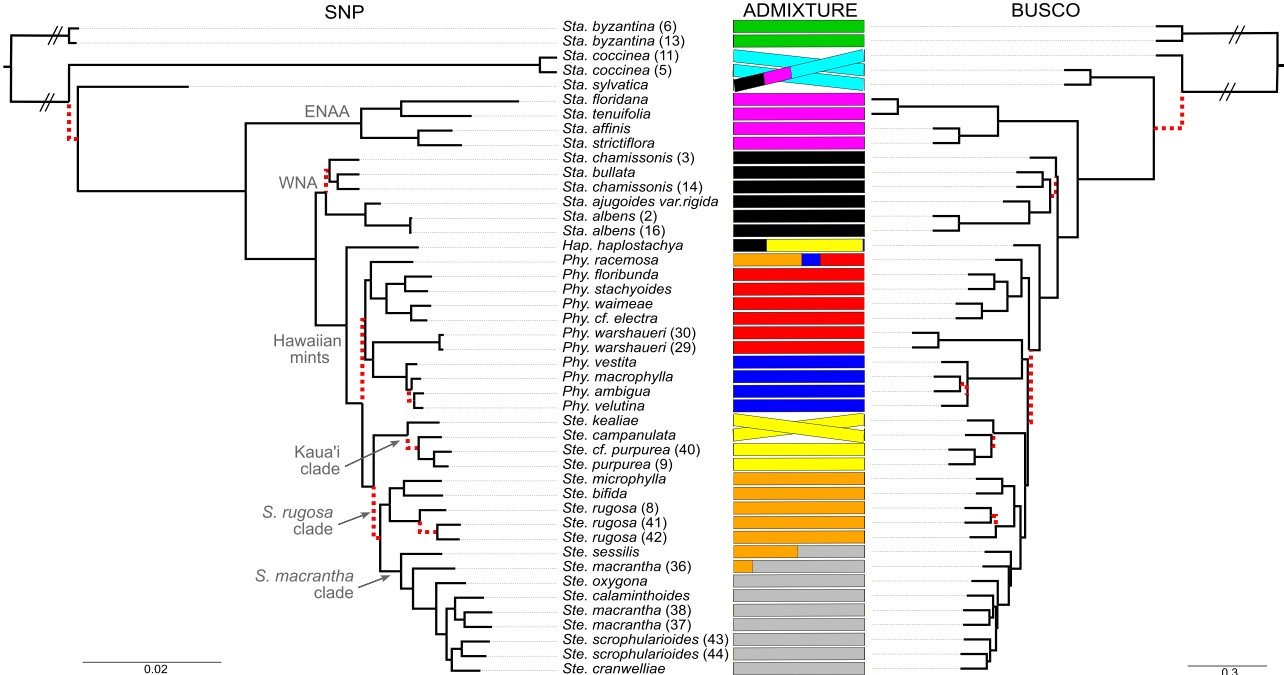

**Fig. 2 | Phylogenetic relationships and admixture among Hawaiian mints and relatives.** The phylogenetic trees are based on maximum likelihood analysis of single nucleotide polymorphism (SNP) dataset DS4 (left; see also Supplementary Fig. 7) and an ASTRAL coalescence tree of BUSCO single copy nuclear genes (right; see also Supplementary Fig. 10). Clade names used in the text are indicated on the SNP tree: ENAA (eastern North American and Asian *Stachys*), WNA (western North American *Stachys*). *Sta.* = *Stachys*, *Hap.* = *Haplostachys*, *Phy.* = *Phyllostegia*, *Ste.* = *Stenogyne*; numbers after taxon names refer to project ID number for taxa that have more than one individual represented (see Supplementary Data 4). Discordance between the two trees is marked with a tanglegram and dotted, red-colored branches and crossed bars in the AMIXTURE plot. ADMIXTURE results shown between the two trees with each color representing a separate ancestral population, *K* (best-fitting *K* = 9 is shown). Source data are provided as a Source Data file.

*Phyllostegia* was resolved as paraphyletic (Fig. 2, Supplementary Fig. 10). An additional discordance between the SNP- and BUSCO-based trees was that the position of *Stachys sylvatica* and *Stachys coccinea* interchanged. In cases of discordance within Hawaiian mints, differences corresponded to very short internal branches in both trees, possibly caused by rapid radiation bursts that are difficult to tease apart phylogenetically. There was also discordance within specific lineages, such as the *Stenogyne* Kaua'i clade, wherein *Stenogyne kealiae* (in the SNP tree), instead of *Stenogyne campanulata* (in the BUSCO tree), occupied a position as sister to the rest of this clade. There was one instance of discordance within WNA *Stachys*s, where one *Stachys chamissonis* individual was sister to all other WNA *Stachys* samples plus Hawaiian mints in the BUSCO tree only; this sample instead grouped within WNA *Stachys* in the SNP tree. Such differences based on coalescent analyses of single loci versus a whole-genome variant average may reflect ILS being particularly confounding in the heritage of this lineage.

## Whole genome duplication histories reveal four independent polyploidy events

To evaluate polyploid depths among species, we compared percent duplicated BUSCOs (D) in the de novo Illumina assemblies (Fig. 3a, Supplementary Fig. 9). D scores > 35% marked likely polyploid lineages, whereas scores of ~4% suggested diploid status. There were three clades of putative polyploids, the Hawaiian mints and their closest WNA *Stachys* relatives, the *Stachys coccinea* lineage, and the *Stachys sylvatica* lineage, whereas the outgroup taxon *Stachys byzantina* and most ENAA *Stachys* taxa (*Stachys floridana*, *S. tenuifolia*, and *S. strictiflora*) appeared to be diploid. These determinations were also reflected by chromosome counts. *Stachys byzantina* and *Stachys floridana* (and diploid populations of *Stachys tenuifolia*) are $2n = 30$ and 34, respectively[44,45]. This number is roughly doubled in Hawaiian mints (e.g., *Stenogyne purpurea*, $2n = 66$), WNA *Stachys* relatives (e.g., *Stachys bullata*, $2n = 66$[18]), and *Stachys sylvatica*, which is $2n = 66$[44]. *Stachys coccinea* differs with $2n = 84$[44], suggesting that this species has a more complicated polyploid history. One ENAA *Stachys* sample, *S. affinis*, had an excess of duplicated BUSCOs, with about three times the number in *Stachys byzantina*, but not to the same level as the other paleo-octoploids. However, chromosome counts of this sample report $2n = 66$[18], suggesting that it has also experienced a WGD event. Because this WGD is not shared with other members of its clade (other ENAA taxa) and these other apparent high-polyploid taxa are interspersed within *Stachys sensu lato* phylogeny, it is probable that each WGD observed represents an independent event (Supplementary Fig. 9).

Next, to further evaluate WGD independence and investigate subgenome sharing and potential progenitor lineages of the WGD among Hawaiian mints and their closest relatives, we used the GRAMPA[46] application to resolve a subgenome phylogeny from multicopy BUSCO genes (Supplementary Fig. 11). This analysis showed that the Hawaiian endemic mints and WNA *Stachys* likely shared an allopolyploidy event, with a diploid ancestor related to *Stachys coccinea* (hereafter refered to as "UC" – unsampled *Stachys coccinea*-like) contributing one progenitor lineage, and a diploid ENAA *Stachys*-like ancestor contributing the other. Based on a previous study of five low-copy, independently inherited nuclear loci[19], this UC ancestor could represent a relative of unsampled Mesoamerican or South American species, such as Mesoamerican *Stachys agraria* or South American *Stachys eriantha* (both $2n = 32$[44,47]), which belong to the "Meso-SA I" clade. *Stachys coccinea* is found in their "Meso-SA II" clade[19]. GRAMPA also resolved *Stachys sylvatica* to be an allopolyploid of distinct origin, involving a relative of *Stachys coccinea* and a *Stachys byzantina*-like ancestor, despite its similar chromosome number to Hawaiian mints. Finally, *Stachys coccinea* itself was resolved as a third, independent allopolyploid (and likely allohexaploid) lineage also descendant from the ENAA *Stachys* lineage and an unsampled, closely related ancestor,

which may be a similar but not necessarily the same progenitor inferred for the WNA *Stachys*/Hawaiian mint hybridization event (Supplementary Fig. 11). Lastly, GRAMPA placed both *Stachys affinis* subgenomes in the same clade with its sister taxon, *Stachys strictiflora*, indicating a rather close and recent allopolyploidy, or perhaps an autopolyploid event (Supplementary Fig. 11). These predicted and independent allopolyploidy events are summarized in Fig. 3a.

Overall, polyploidy events appear to be recurrent in the *Stachys s.l.* lineage[26], but they are not necessarily paired with phylogenetic radiations. Likewise, numerous plant radiations do not seem to be associated with polyploidy events, such as Hawaiian *Melicope*[48]. However, it is still possible that the polyploid event underlying Hawaiian mints and North American *Stachys* contributed some genomic substrate for evolutionary radiation of the lineage in the dynamic and fast-evolving Hawaiian landscape[49].

## Genome phasing of *Stenogyne calaminthoides* supports the ENAA and UC diploid lineages as progenitors for the Hawaiian mints and WNA relatives

We uncovered numerous differentiating features unique to either a homeologous chromosome of ENAA origin, versus one of UC origin. First, when calculating depth of coverage across chromosomes, eastern North America *Stachys* species (*Stachys floridana* and *S. tenuifolia*) had a set of 17 chromosomes that mapped with high coverage and 15 chromosomes that mapped with much lower coverage (Supplementary Fig. 12, Supplementary Data 9). Conversely, the 15 chromosomes that mapped with lower coverage in ENAA samples mapped with high coverage in *Stachys coccinea*, while the 17 chromosomes with high coverage in ENAA instead had low coverage in *Stachys coccinea*. These differences are suggestive of ENAA and UC being the progenitor lineages for *Stenogyne calaminthoides* (and by extension, all Hawaiian mints and WNA *Stachys*; Fig. 3a). The 17 chromosomes belonging to the ENAA subgenome likely have higher depth of coverage with ENAA samples due to close phylogenetic relationship, mapping poorly to the UC chromosomes due to their more distant phylogenetic relationship. We found a few unique cases in which the pattern was violated, i.e., where we saw roughly twice the expected coverage for a specific chromosome. For example, sister taxa *Phyllostegia floribunda* and *P. stachyoides* shared doubling of an UC chromosome, and the two *P. warshaueri* specimens shared doubling of a different UC chromosome (Supplementary Fig. 12). All of these instances reflect different known chromosome numbers, and hence, independent formation events. Interestingly, *Stenogyne* species had no such chromosomal variation detected, consistent with known chromosome counts $2n = 64$, while *Phyllostegia* chromosome counts vary, with $2n = 64$ or 66[31]. The latter appear to be cases of aneuploidy, specifically gain of a single chromosome, which has been thoroughly studied in the case of the allopolyploid composite *Tragopogon*[50].

We were further able to identify homeologous pairs of chromosomes, each consisting of one ENAA chromosome and one UC chromosome, using both SubPhaser[51] and DAGchainer[52] in CoGe (Supplementary Data 10). SubPhaser clearly split the ENAA and UC subgenomes based on *k*-mers, the former into a monomorphic cluster suggesting highly similar repeat content, while the latter was much more polymorphic, possibly because UC represents an unsampled lineage that was defined on the basis of the allopolyploid *Stachys coccinea*, or that the UC lineage has a deeper coalescence, based on the phylogeny of chromosomes (Fig. 3b, Supplementary Fig. 13). Subgenome-wise dating of LTR blooms using two different mutation rate estimates (6.7E-09 vs. 1.75E-09 mutations per site per year, emulating *Arabidopsis*[53]) yielded medians of ~4.5 Million years ago (Mya) and 17 Mya, respectively. The younger date closely corresponds with a previous estimate for the Hawaiian mints/North American *Stachys* clade origin based on nuclear ribosomal and plastid genome markers[18].

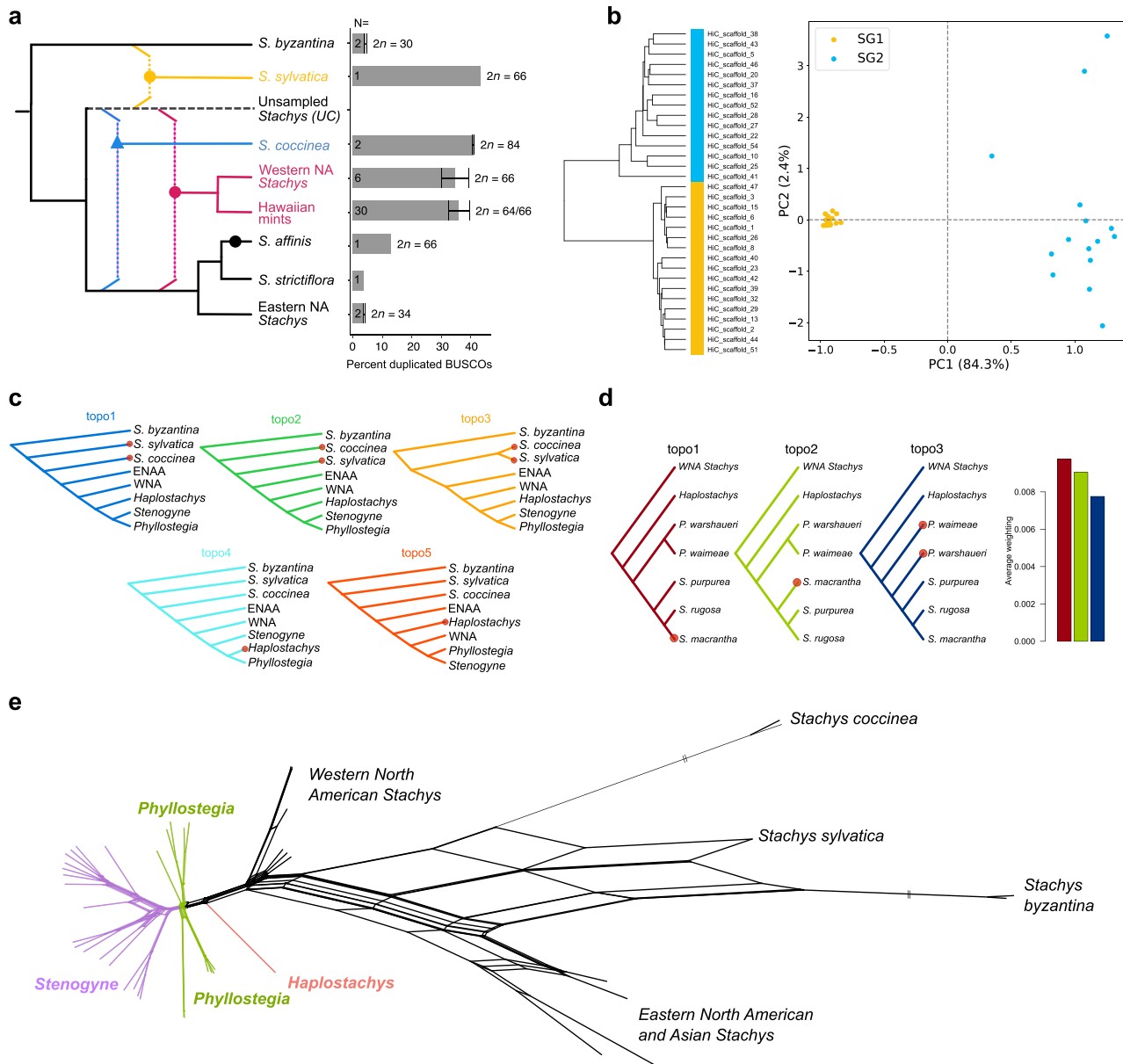

**Fig. 3 | Allopolyploid history and phylogenetic conflict among Hawaiian mints and relatives. a** Left panel: Schematic tree showing hypothesized allopolyploid events among sampled taxa (see text for details). Circles represent polyploid events and the triangle indicates a potential hexaploidy event. Dotted lines associated with colored symbols indicate progenitor lineages of the allopolyploidy events, while the dashed branch leading to "unsampled *Stachys* (UC)" represents a hypothetical, unsampled diploid relative of *Stachys coccinea*. Timing and order of polyploidy events are not to scale. Right panel: The mean percentages of duplicated BUSCO genes and known chromosome counts (2*n*) for each group are displayed in the schematic tree. Data and N are presented as mean values ±SD and number of individuals, respectively. **b** PCA of kmers from reference *Stenogyne* *calaminthoides* inferred from SubPhaser. SG1 represents the ENAA subgenome and SG2 represents the UC subgenome. **c** The top five topologies found from *Twisst* analyses of dataset DS4 (all taxa included). Taxa that move among trees are highlighted with red dots. See Supplementary Fig. 15a for details of the groupings used. **d** The top three trees found from *Twisst* analyses based only on Hawaiian mints and WNA *Stachys* relatives. Taxa that move among trees are highlighted with red dots. The bar plot shows the average weighting values for each tree. See Supplementary Fig. 15b for details of the groupings used. **e** Phylogenetic network (NeighborNet) of all samples (dataset DS4). See Supplementary Fig. 17 for complete network with branches labelled. Source data are provided as a Source Data file.

Using CoGe[54], we output syntenic gene pair results to a Circos plot, which also demonstrated relatively clear pairings between homeologs (Fig. 1b), with the exception of two ENAA chromosomes that did not have a clear syntelog among chromosomes greater than 25 Mb. Indeed, also confirmed using CoGe SynMap[55] (Supplementary Fig. 14), chromosome 47 seemingly shared synteny with blocks smaller than 25 Mb, possibly representing scaffolding errors or potentially chromosomal fission events. The other singleton, chromosome 42, was partially syntenic to smaller scaffolds and partially syntenic to another chromosome (chromosome 41), possibly also representing

scaffolding errors or chromosomal fusions. Hence, for downstream analyses based on chromosome and subgenome comparisons, we omitted these chromosomes of unclear homeology focusing instead on the 15 well-defined syntenic pairs.

## Incongruent phylogenetic signal among the Hawaiian mints and relatives

Since the Hawaiian mint radiation likely involved divergence over a relatively short time period, extensive ILS may be a confounding factor for phylogenetic reconstruction. We aimed to search for incongruent

phylogenetic signals along the genome using *Twisst*[56] (topology weighting by iterative sampling of sub-trees). First, to investigate competing topologies, each Hawaiian mint genus was assigned a group and *Stachys* was separated into five monophyletic groups (Supplementary Fig. 15a), producing 25,656 SNP trees based on windows of 50 SNPs each. We found that the dominant topology (Fig. 3c) followed the BUSCO tree topology, followed by the genome-wide average SNP tree topology (Fig. 2). In the third most common topology, *Stachys sylvatica* and *Stachys coccinea* were resolved as sister taxa. Indeed, most of the conflicting phylogenetic signals among the 50 SNP windows could be attributed to different positioning of *Stachys sylvatica* and *Stachys coccinea*. This was expected given our model of subgenome sharing, where *Stachys sylvatica* and *Stachys coccinea* both possess at least one subgenome from the lineage leading to UC *Stachys* (Fig. 3a).

Next, to tease apart phylogenetic signals by subgenome, we plotted *Twisst* results according to chromosome assigned to each of the ENAA and UC subgenomes. We found a clear trend in phylogenetic signal for the UC chromosomes compared to the ENAA chromosomes (Supplementary Fig. 16). The ENAA chromosome topologies were fairly consistent with the pattern seen using genomewide SNPs, with the two top topologies mainly reflecting *Stachys sylvatica* and *Stachys coccinea* phylogenetic discordance (Supplementary Fig. 16a). For four of the fifteen ENAA chromosomes, beginning with the third best topology, phylogenetic placements within Hawaiian mints shuffled, with *Stenogyne* and *Haplostachys* switching places. Subsequent best trees had *Haplostachys* and WNA *Stachys* interchanging positions. Both of these cases are consistent with the fourth- and fifth-best trees resulting from all genomic windows (Fig. 3c). For the UC chromosomes (Supplementary Fig. 16b), the topology weights were roughly between one-half to one-third that of the ENAA chromosomes, matching the SubPhaser repeats discordance pattern (Supplementary Fig. 13).

As there was some shuffling observed among the Hawaiian mints, as described above, we next performed *Twisst* based on only Hawaiian mints and their WNA *Stachys* relatives, all expected to be descendants of the same allopolyploidy event, and hence, contain the same subgenomes (see Supplementary Fig. 15b for the groupings used). We used *Twisst* with 16,652 SNP trees, each based on non-overlapping windows of 50 SNPs. Again, we found that the dominant, most represented tree was consistent with the BUSCO and SNP trees (Fig. 3d). In the second most represented tree, monophyly of each Hawaiian mint genus was retained, although the *Stenogyne* Kaua'i clade (indicated by *Stenogyne purpurea*) grouped with the *Stenogyne rugosa* clade instead of being sister to all other *Stenogyne*, similar to the BUSCO tree (Fig. 2). However, in the third-best topology, monophyly was violated, in that *Phyllostegia* became paraphyletic, consistent with the BUSCO tree.

To further explore patterns of phylogenetic conflict, we also generated a NeighborNet[57]. We uncovered a five-pointed "star" of ambiguity among *Stachys byzantina*, *S. sylvatica*, *S. coccinea*, ENAA *Stachys*, and WNA *Stachys* plus Hawaiian mints (Fig. 3e, Supplementary Fig. 17), edges within which are nonetheless consistent with the patterns of the hypothesized subgenome sharing. For example, *Stachys sylvatica* lies between *Stachys coccinea* and *Stachys byzantina* in the star, supporting it as an allopolyploid containing both *Stachys byzantina*-like and UC subgenomes. We also generated a NeighborNet using only Hawaiian mints to better identify conflicting signal within that lineage alone (Supplementary Fig. 17b). Supporting some of the *Twisst* topologies, *Phyllostegia* is paraphyletic in the NeighborNet, unlike *Stenogyne*, which despite the phylogenetic conflict within the genus is resolved as a monophyletic group. Finally, we used DensiTree[58] to further explore relationships among the Hawaiian genera and closest North American outgroups using subsampling of SNPs by linkage group, in one case via whole-chromosome trees, and in the second by genomic windows of 25 Kb. These analyses similarly visualized significant phylogenetic incongruence among taxa (Supplementary

Fig. 18). Interestingly, in some trees *Haplostachys* grouped with a subclade of *Phyllostegia*, which in some cases showed incongruent relationships to a second *Phyllostegia* subclade. Within *Stenogyne*, incongruent relationships between the Kaua'i and *Stenogyne rugosa* clades were revealed.

## The Hawaiian mint genera display contrasting genetic structure and diversity

To further explore patterns of relationship among Hawaiian mints and close *Stachys* relatives, we used principal component analysis (PCA)[59] based on SNP data. As expected, most of the variation in the dataset including all the taxa (DS4) was explained by differences among the distant *Stachys* outgroups (Supplementary Fig. 19a). The first two principal components (PC1 and PC2) primarily distinguished *Stachys byzantina* and *Stachys coccinea*, respectively. The next PC (PC3) mainly separated ENAA species from the rest. PC2 and PC3 could represent SNP diversity unique to subgenomes or ploidy levels, with PC2 representing UC and PC3 representing ENAA; indeed, *Stachys sylvatica*, which shares some of each subgenome, placed between the ENAA and *Stachys coccinea* and the "ingroup" taxa, respectively, in these two plots. To gain resolution for ingroup taxa, we subsequently removed *Stachys byzantina* and found that a rough cline appeared among the remaining *Stachys* taxa and the Hawaiian mints, one that is especially linear for Hawaiian mints and their close WNA *Stachys* relatives (Fig. 4a, Supplementary Fig. 19b). Such clinal variation may correspond to progressive cladogenesis via geographic speciation[22]. Among the first four PCs identified, *Stachys chamissonis* and *Stachys bullata* were the taxa most proximate to the Hawaiian mints. Next, to more deeply analyze diversity among the Hawaiian mints, we removed *Stachys* from the PCA analysis (Fig. 4a, Supplementary Fig. 19c). Here, *Haplostachys* was separated from the remaining Hawaiian mints in the first PC, as expected given its long branch in phylogenies and its large number of singleton/doubleton SNPs (see next paragraph). Further, in PC2, *Phyllostegia* remains a tighter cluster, whereas *Stenogyne* segregated into clusters by clade identity, including the *Stenogyne macrantha* clade, the *Stenogyne rugosa* clade, and the Kaua'i clade, the latter grouping closest to *Phyllostegia*. *Stenogyne* formed a cline along PC3, and in PC4 *Phyllostegia* largely separated into three clusters. When plotting PC2 and PC3 together and connecting the samples to the phylogenetic tree based on SNP data, the *Stenogyne* cline show a nearly perfect phylogenetic order (Fig. 4a). Further, the *Phyllostegia warshaueri* samples shared a long branch in the SNP phylogeny, and these samples were distinct in the PCA as well. Although most samples followed a pattern consistent with phylogenetic progression, some samples appeared to be unique, such as *Phyllostegia racemosa*, which tends to cluster within or near *Stenogyne*, pointing to an admixed origin (Supplementary Fig. 19c; also see below discussion). The clinal variation observed among the Hawaiian mints may correspond to progressive cladogenesis via geographic speciation[22]. For example, allelic variation may have become fixed along an ongoing cladogenetic process caused by serial founder events in an island-hopping model of geographic speciation[60]. Such relatively simple isolation-by-distance processes may be facilitated in an extremely young and rapidly expanding and dissecting volcanic environment, such as that of the Hawaiian Islands.

We further investigated the diversity among the mint genomes using various population genetic statistics. Of note, we found that *Haplostachys haplostachya* was highly unique in that it had significantly higher number of private alleles than any of the other Hawaiian mints (Fig. 4b) and in fact had the most doubletons (homozygous sites for a unique SNP) and the second highest level of singletons (*Stachys sylvatica* had the most singletons) among all taxa (Supplementary Fig. 20a). Assuming a single colonization event of the Hawaiian lineage, the *Haplostachys* doubletons are likely rather recent, having evolved since the split of *Haplostachys* from the rest of the

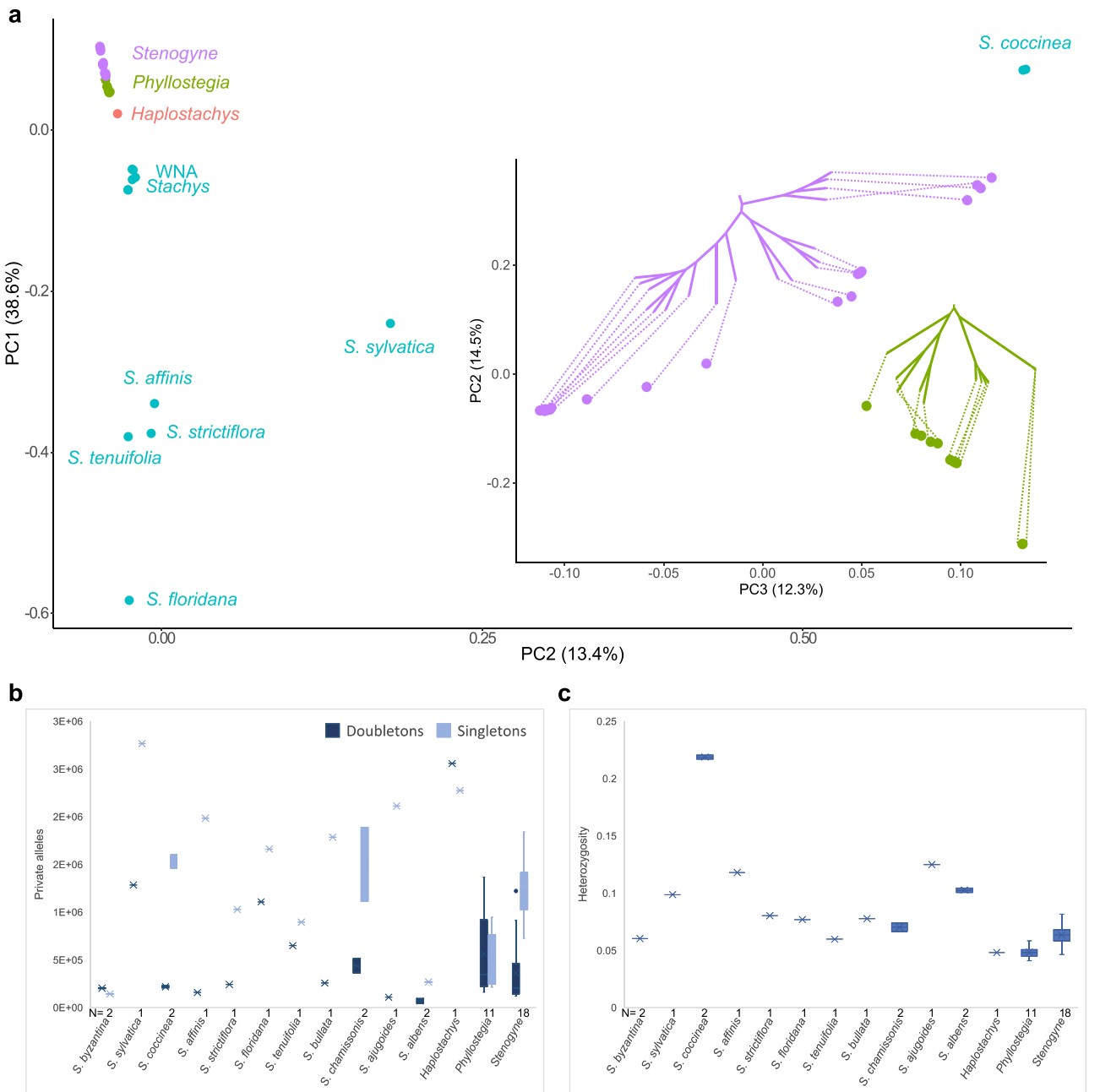

**Fig. 4 | Genetic structure and diversity among Hawaiian mints and relatives.**
**a** Principal component analysis (PCA) of dataset DS4c (*Stachys byzantina* excluded). The insert shows a PCA plot of *Stenogyne* (purple) and *Phyllostegia* (green) with dotted lines connecting them to the phylogenetic tree based on SNP data.
**b** Singletons and doubletons (homozygous sites for a unique SNP) and (**c**) Heterozygosity based on SNP dataset DS7 (Supplementary Data 6). Individuals for each Hawaiian genus, respectively, and representatives of the same *Stachys* species were combined. Box plots depict the median (center line), the mean (cross), and the upper and lower quartile box limits (whiskers are the first and third quartiles). Number of individuals (N) are shown below the X-axis. Source data are provided as a Source Data file.

Hawaiian mints, given that no other Hawaiian mint shares these alleles. An increase of rare alleles could also be associated with an ancient range expansion[61], which is consistent with the former wide distribution of *Haplostachys* across three islands (Hawaiʻi, Kauaʻi, and Maui), although *Haplostachys* is today only found on Hawaiʻi[24]. An alternative hypothesis is that the Hawaiian Islands were colonized in two events, involving slightly different hybrid-polyploid lineages (as discussed above), as the number of private alleles can increase with age of the allopolyploid[41], suggesting that *Haplostachys* may represent an older sibling allopolyploidy event. Interestingly, the number of heterozygous sites for *Haplostachys* is comparable to other Hawaiian mints that in general have lower levels of heterozygosity than their mainland relatives, although *Stenogyne* species tend to have more heterozygous sites than their Hawaiian relatives (Fig. 4c, Supplementary Fig. 20b). These results suggest that closely related Hawaiian taxa, including *Phyllostegia* and *Stenogyne*, which both experienced extensive morphological and rapid diversification, may exhibit contrasting genetic structure and diversity levels, likely caused by different paths of radiation and subsequent isolation.

**The majority of present-day Hawaiian mints appear to be unadmixed**
To explicitly test for possible signatures of admixture among our Hawaiian mint samples, we employed a multifaceted approach using

the $f_3$-statistic[62] to compare all possible 3-way combinations of samples, as well as TreeMix[63] and ADMIXTURE[64]. Our results show that significantly negative $f_3$ values reflected close phylogenetic relationships among samples, as anticipated based on previous studies[16]. Crosses of negative values in our $f_3$ heat maps (wherein all source combinations showed negative $f_3$) likely reflected strong identity by descent (IBD), as also seen in other organisms, such as *Syzygium*[22] and *Ursus*[65]. For example, the two representatives of *Phyllostegia warshaueri* was marked by the cross pattern of negative values when one was configured as a target and the other as a source (Supplementary Fig. 21). We then employed TreeMix, for which a migration edge was observed between *Stenogyne sessilis* and a *Stenogyne rugosa* relative (Supplementary Fig. 22). *Stenogyne sessilis* also appeared to be admixed in the ADMIXTURE analysis (best-fitting $K = 9$; Fig. 2), and it appears consistently admixed across most $K$ values (Supplementary Fig. 23). This taxon also showed extensive webbing involving *Stenogyne sessilis* and *Stenogyne rugosa* in the NeighborNet analysis (Supplementary Fig. 17), which further suggests the former taxon may be admixed. Another possibly admixed species is *Phyllostegia racemosa*, which in ADMIXTURE results was resolved as a mixture of *Phyllostegia* clades and the *Stenogyne rugosa* clade. Although this admixture signal was not identified by TreeMix or $f_3$-statistics, this taxon clustered closely with *Stenogyne* in the PCA (Supplementary Fig. 19c). It is also possible this could represent shared, incompletely sorted ancestral alleles. As such, it is not clear if *Phyllostegia racemosa* represents a case of ILS or gene flow. Similar to *Phyllostegia racemosa*, one *Stenogyne macrantha* individual appeared admixed with ADMIXTURE analysis (Fig. 2), but there was no strong support for this in other analyses. Intriguingly, ADMIXTURE showed *Haplostachys haplostachya* as sharing ancestral groups with the *Stenogyne* Kaua'i clade ($\sim$74%) and WNA *Stachys* (~25%), and only a small fraction with *Phyllostegia* (~1%). Such patterns, however, were not identified by $f_3$ or TreeMix as stemming from inter-lineage admixture, and could instead reflect ILS among WNA *Stachys* and early- and rapidly-diverging Hawaiian mint clades. It is important to note that ADMIXTURE results are often overinterpreted as indicative of cross-lineage admixture[66], and the $K$ components from ADMIXTURE simply represent subsets of inherited SNP variation that could reflect any underlying mixtures, of which ILS may be another underlying causal factor[22].

Hybridization has been identified as one key contributor to radiations[67], and the Hawaiian mints appear to be no exception. However, the principal hybridization detected, an allopolyploidy event, occurred prior to colonization of Hawaii and subsequent diversification, with only limited interspecific gene flow occurring among the Hawaiian taxa. Ancient hybridization, in addition to polyploidization, may instead have contributed to a rich genomic diversity among Hawaiian mint ancestors, possibly facilitating radiation in the context of ecological opportunity and founder effects.

**Admixture and demographic history in a putative hybrid swarm**
Because signals of ancient interlineage admixture may be confounded by ILS, we finally sought to explore admixture in recent times as a potential source of diversification in the Hawaiian lineage. We investigated a putative hybrid swarm of *Stenogyne* individuals found on the Mauna Kea volcano on the island of Hawai'i. This population has been predicted to comprise largely F1 hybrids with frequent backcrossing into *Stenogyne microphylla*, featuring a spectrum of morphological traits intermediate between *Stenogyne microphylla* and *Stenogyne rugosa*[24]. Generally, *Stenogyne microphylla* is known for its small leaves and tendency to grow as vines in *Sophora chrysophylla* trees, while *Stenogyne rugosa* usually grows in the shade below these trees, making the latter susceptible to feral ungulate grazing on the slopes of Mauna Kea[16]. Ten *Stenogyne* individuals each were collected in 11 sites along the Kaaliali trail on Mauna Kea (Fig. 5a), and the 109 individuals were resequenced, mapped to the *Stenogyne calaminthoides* reference

genome and SNPs were called as we did for other species in this study (Supplementary Data 11). We included four additional samples in our analyses: three presumed unadmixed representatives (on the basis of morphology and geographic location) of *Stenogyne rugosa*, and one ostensibly unadmixed *Stenogyne microphylla*. We first generated a phylogenetic network to visualize potential conflicting phylogenetic signal in the data (Supplementary Fig. 24). The majority of the samples group with the putatively unadmixed *Stenogyne microphylla* in a tight cluster, suggesting common ancestry. However, the branches leading from this cluster to the putatively unadmixed *Stenogyne rugosa* feature extensive webbing and represent individuals primarily from site 0.1 and site 1.0, and to a lesser extent, site 0.2, at the two ends along the trail. Most of site 0.1 samples form a webbed lineage leading directly to *Stenogyne rugosa*, while site 1.0 samples appear distinct, suggesting that individuals at this location share alleles with an unsampled species. To test if a different *Stenogyne* species could fall within this group, we added all our *Stenogyne* species to the analysis and found that none of our sampled species cluster within or near site 1.0 samples (Supplementary Fig. 25). These results were consistent with maximum likelihood phylogenetic tree analysis of the Mauna Kea samples, in which a clade consisting of samples from site 1.0 was distinct from *Stenogyne rugosa* (Supplementary Fig. 26). A previous study based on amplified fragment length polymorphism (AFLP) fragment data found that *Stenogyne rugosa* samples from Mauna Kea grouped with a sympatric species, *Stenogyne angustifolia*[16], which could represent the unsampled taxon in this case.

To further explore these intial findings, we applied analyses of ADMIXTURE and PCA. ADMIXTURE indicated that most samples (site 0.4 through 0.7) represented unadmixed *Stenogyne microphylla*, while samples with admixed ancestry were found primarily in sites 0.1 and 1.0 (Fig. 5b, Supplementary Fig. 27). Samples from sites occurring next to these latter locations (0.2, 0.9, and 1.1) also showed some *Stenogyne rugosa* ancestry. In PCA (Supplementary Fig. 28), PC1 mainly separated two *Stenogyne rugosa* accessions from South Kona Forest Reserve that are recognized by having uniquely tomentose leaves; otherwise, most principal components exhibited a tight cluster comprised of *Stenogyne microphylla* and individuals from most of the collected sites. In addition, distinct clines involving samples from mainly two sites emerged (Fig. 5c): one cline consisting of site 0.1 and 0.2 samples leading to *Stenogyne rugosa* (sample #8, collected from Mauna Kea) and the second cline consisting of only 1.0 samples. Next, we employed the $f_3$-statistics of selected samples as target and all *Stenogyne* samples as potential sources to compare individuals that appeared admixed to those that appeared relatively unadmixed with ADMIXTURE (see Supplementary Fig. 27). Two hypothetically admixed samples (from site 0.1 and 1.0, respectively) exhibited particularly strong patterns of identity by descent (IBD) with other presumed admixed individuals from the same sites, suggesting localized allele sharing, apparently also involving *Stenogyne rugosa* (Supplementary Fig. 29a, b). One presumably unadmixed individual from site 0.6 showed strong IBD with other individuals of suggested *Stenogyne microphylla* ancestry (Supplementary Fig. 29c), while the apparent unadmixed individual from site 0.1 showed strong IBD with the three presumably unadmixed *Stenogyne rugosa* samples but otherwise exhibited largely positive $f_3$ values (Supplementary Fig. 29d). Increased heterozygosity and private alleles was found among individuals from the 0.1 and 1.0 sites (Supplementary Fig. 30), which likely reflects increased allelic diversity due to introgressive hybridization.

Finally, to test for an association between the observed patterns of genetic structure with morphology, we recorded the following traits for all samples of *Stenogyne microphylla* and *Stenogyne rugosa* and their purported hybrids: trichome length, stem width, average leaf area, and average width and length ratios across three leaves for each individual (Supplementary Data 12). We tested for potential correlation among these morphological traits and principal components based on

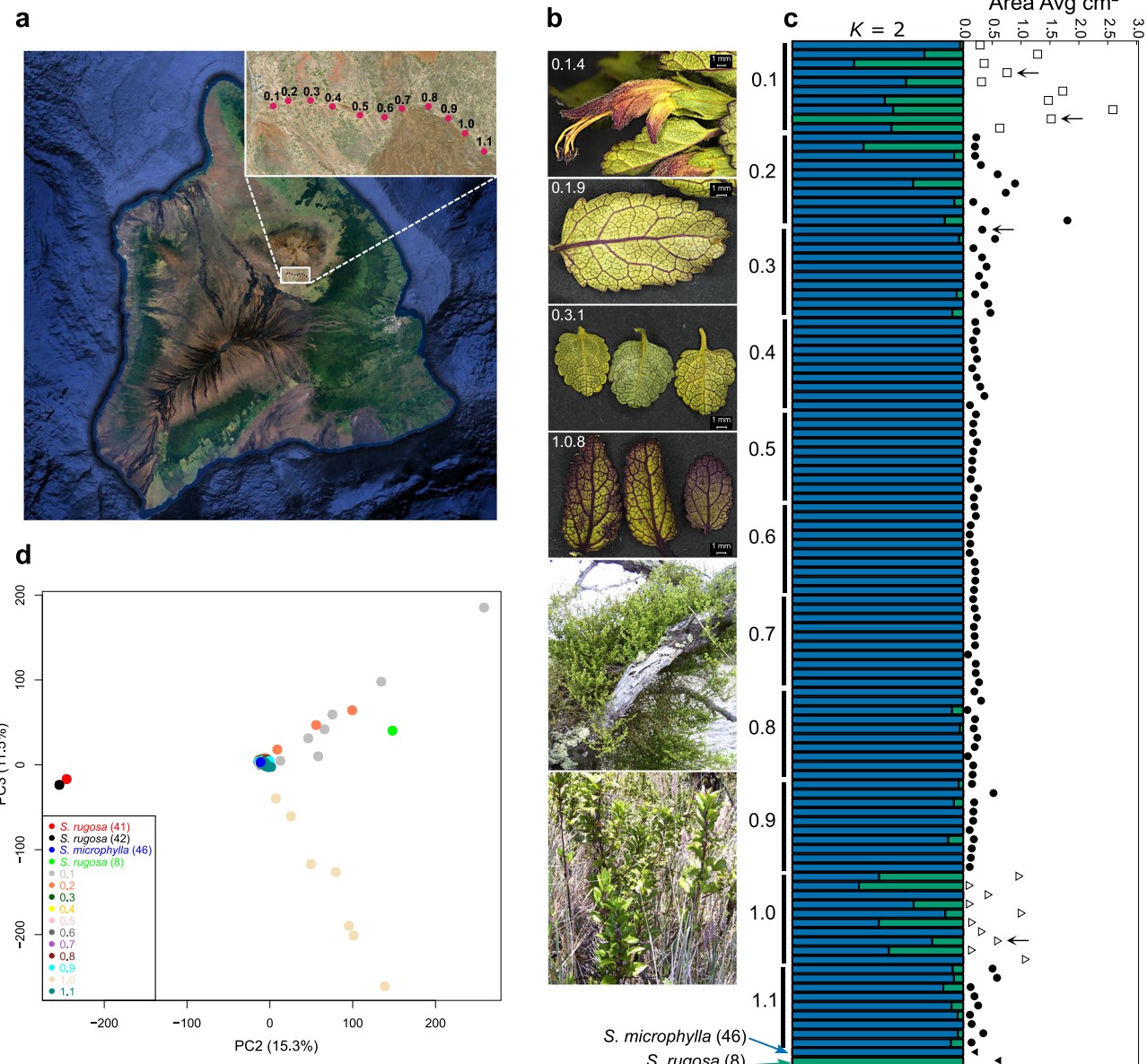

**Fig. 5 | Admixture and demographic history in a putative hybrid swarm of *Stenogyne microphylla* and *Stenogyne rugosa*.** **a** Map of Hawaiʻi showing the location of the hybrid populations on Mauna Kea (red star), with the Kaaliali trail zoomed in. The dots represent the 11 sites sampled along the Kaaliali trail, each with ten *Stenogyne microphylla* and purported hybrid individuals collected and analyzed. **b** Silica-gel-preserved plant material and habitat images of *Stenogyne* individuals from Mauna Kea, labeled by site where relevant. Top to bottom: 0.1.4, an open flower; 0.1.9, a somewhat elongate, *S. rugosa*-like leaf; 0.3.1, smaller, *S. microphylla*-like leaves; 1.0.8, elongate to short leaves, representing an admixed individual; *S. microphylla*-like plant climbing on a *Sophora* tree; *S. rugosa*-like plant, entirely terrestrial in habit. **c** ADMIXTURE results (dataset HM1), with best-fitting

*K* = 2, is shown to the left (numbers refer to the sampling sites shown in (**a**), and the average leaf area measured from each individual is displayed to the right. Open squares and triangles refer to the mostly admixed samples from sites 0.1 and 1.0, respectively, while the filled circles and triangles indicate mostly unadmixed and the *S. microphylla* and *S. rugosa* reference samples, respectively. Arrows correspond to the samples imaged in (**b**) (also from top to bottom). **d** Principal component analysis (dataset HM1), showing PC2 and PC3. Source data are provided as a Source Data file. Image sources for panel **a** are Google Earth, Image Landsat / Copernicus, Data LDEO-Columbia, NSF, NOAA, Data SIO, NOAA, U.S. Navy, NGA, GEBCO, Data MBARI; inset, web map service data provided by USDA.

SNP data and admixture proportion from ADMIXTURE analysis (Supplementary Fig. 31). Overall, we found only weak correlation between genetic data and morphological traits. Leaf area is one of the main distinguishing features between *Stenogyne microphylla* and *Stenogyne rugosa*, with *S. microphylla* having far smaller leaves[24]. The strongest correlation between morphology and the genetic data was indeed between PC1 and average leaf area (Supplementary Fig. 32). Accordingly, the majority of samples, particularly from sites 0.3 through 0.9, had small leaves, except for the site 0.1–0.2 and 1.0 individuals, where samples generally had larger leaves, with some even larger than the

unadmixed *Stenogyne rugosa* individual (Fig. 5c), supporting their hybrid background.

In summary, the genetic and morphological data point to recent, albeit localized and limited, admixture on Mauna Kea between *Stenogyne microphylla* and *Stenogyne rugosa*, and possibly also an unsampled taxon related to *Stenogyne angustifolia*. However, these events may well be historical and anthropogenic in origin, related to introduced ungulate herbivory, and therefore not reflective of any ancient admixture processes. It is also possible that such introgression events between closely related taxa, via limited breakdown of recent

parapatry, have commonly occurred during early diversification of different Hawaiian mint lineages.

## Hawaiian mints provide key insights on island plant radiations

In this study we have uncovered the most detailed evolutionary history to-date of a major Hawaiian plant radiation, the endemic mints, generating a near chromosome-level reference genome and whole-genome analyses of 23 Hawaiian mint species and 11 of their Old and New World relatives, in addition to over 100 individuals of two *Stenogyne* species, and their purported hybrids, found on the Mauna Kea volcano. We confirmed that the Hawaiian mints are monophyletic and most closely related to western North American *Stachys*. However, contrary to our initial expectations[16], given their hypothesized hybrid origin, our analyses demonstrate that Hawaiian mints do not appear to be highly admixed at present, except for localized introgressive hybridization on Mauna Kea. Instead, the phylogenomic incongruence we observed more likely reflects a combination of distinct subgenome evolutionary histories and/or ILS. Hawaiian mint genomes have clearly been duplicated twice since common ancestry with grapevine, with one ancient WGD shared among most Lamiales, and a more recent WGD shared between Hawaiian mints and their closest western North American *Stachys* relatives. We found strong support that this latter, most recent WGD was an allopolyploidy event, likely involving hybridization of a relative of the eastern North American and Asian *Stachys* and an unsampled diploid *Stachys* lineage related to *Stachys coccinea*. Additionally, we discovered that independent WGD events may be prevalent among *Stachys*, with four independent polyploidy events having occurred within our taxon sample. Despite the analytical depth of this study and apparent monophyly of the Hawaiian lineage, the number of times its members may have colonized the Hawaiian Islands remains unclear, with the possibility that the unique genus *Haplostachys* may have had an independent origin (and migration to the Hawaiian Islands) from within the same allopolyploid clade. Indeed, such an additional layer of complexity is hinted at from the excess of *Haplostachys* doubletons. Also supporting sibling allopolyploid events prior to colonization is the interchanging phylogenetic patterns for the two *Stachys chamissonis* individuals between the plastid and nuclear SNP tree, in which only one sample of *Stachys chamissonis* shared the plastid haplotype with *Stachys coccinea* and Hawaiian mints, while other WNA *Stachys* did not. Future work should investigate additional presently unsampled *Stachys* species, particularly those ranging from southern North America into South America, to discover and detail the identity of the other progenitor lineage of the Hawaiian mints, a diploid relative of *Stachys coccinea*, possibly represented today by *Stachys agraria* of the "Meso-SA I" clade.

In summary, our work is consistent with a model of parapatric speciation associated with founder events, concomitant with rapid environmental changes in a dynamic volcanic landscape. The allopolyploid ancestry of the Hawaiian radiation may have set up an extensive underlying genomic diversity that could have fueled morphological distinctions driven by drift alone. However, such rapid evolutionary radiations can set up a "nightmare scenario" for disentangling phylogenetic discordances caused by ILS, wherein allelic inheritance may not follow the cladogenetic sequence of events.

## Methods

### Sample collection, DNA and RNA extraction, and genome sequencing

Young leaf tissue was removed from a cultivated individual of *Stenogyne calaminthoides* and immediately weighed, flash frozen with liquid nitrogen, and placed in a −80C freezer. Roughly five grams of flash frozen tissue was used for high-molecular-weight (HMW) genomic DNA isolation. In order to enrich our extraction for nuclei, we followed the BioNano NIBuffer nuclei isolation protocol[22] in which tissue was ground in liquid nitrogen into a fine powder, then added to 10 mL of 0.2 micron filtered IBTB and incubated for 10 minutes on ice. This mixture was then strained using 100 μm followed by 40 μm filters to remove undissolved plant debris. Triton X-100 (1%) was added to lyse organelles before centrifugation at 2000 x g for 10 minutes to pellet the nuclei. Next, we followed the PacBio shared protocol "Preparing *Arabidopsis* Genomic DNA for Size-Selected -20 Kb SMRTbell™ Libraries", beginning with an addition of 10 mL Carlson Lysis buffer and 25 μL of β-Mercaptoethanol (BME) to the pellet and incubating for two hours at 74 °C, swirling every 30 minutes. Next, we performed a double extraction with chloroform/isoamyl 24:1 using equal volumes, retained the aqueous layer after centrifugation at 3200 rcf for 10 minutes and precipitated the DNA overnight at 4 °C. The sample was then centrifuged for 90 minutes at 3200 rcf, washed with 70% cold ethanol, followed by another centrifugation, and the DNA left to dry for 20 minutes. Genomic DNA was further purified with a Qiagen® Genomic-Tip 500/G as follows: the DNA was resuspended in 20 mL buffer G2 and 20 μL RNase A for a 5 minute room temperature incubation. Next, 100 μL proteinase K was added and incubated for one hour at 50 °C. Finally, the DNA was purified using the QIAGEN Genomic-tip 500/G and the manufacturer's instructions, precipitated overnight at 4 °C with 0.7 volume of isopropanol, and washed twice with 70% cold ethanol. Once the DNA was dried, Tris-EDTA was added and the DNA was placed at 37 °C for two days. The resulting HMW DNA was quantified and quality checked using a Thermo Scientific™ NanoDrop™ Spectrophotometer, a Qubit fluorometer, and agarose gel electrophoresis prior to sequencing. The reference individual was Illumina sequenced to -235 Gb. Oxford Nanopore sequencing was performed on multiple DNA extracts using both a GridION and a PromethION at the J. Craig Venter Institute (La Jolla, CA). All Nanopore sequencing runs were then combined into a single fastq file, resulting in a total of -225 Gb sequence reads.

RNA was extracted using a Qiagen® RNeasy PlantMini Kit for three plant tissues from *Stenogyne calaminthoides*: root (R2), stem (S2) and young leaves (Y3). Library preparation and Illumina-based RNAseq was performed by NovoGeneAIT Singapore.

For Illumina resequencing of all other samples, DNA extraction from 20 milligrams of leaf tissue dried in silica gel was performed using the Qiagen® DNeasy Plant Mini Kit. All procedures followed manufacturer's instructions, except for the final step, in which two elutions of 50 μL each were used in order to increase DNA concentration. Following extraction, DNA was quantitated using a Thermo Scientific™ NanoDrop™ Spectrophotometer. NovaSeq Illumina sequencing was performed by NovoGene AIT in Singapore to -30 Gb per sample for 45 taxa of Hawaiian mints and relatives (Supplementary Data 4) and to -15 Gb for 110 samples of *Stenogyne rugosa* and *S. microphylla* and their purported hybrids (Supplementary Data 11).

### Genome size estimation and initial reference genome assembly and filtering

Illumina reads and *k*-mer based estimation were used to predict the genome size of *Stenogyne calaminthoides*. Jellyfish[68] and KmerGenie[69] predicted a genome size of -1.2 billion bases (Gb) and -1.6 Gb, respectively. KmerGenie was used under both haploid and diploid modes. The haploid report produced a normal concave plot with a clear optimum for a *k*-value at $k = 117$. We also ran KmerGenie under diploid mode, as this mode can distinguish homozygous and heterozygous peaks.

Given the large number of Nanopore raw reads and our computational limitations, the reads were first filtered using NanoFilt[70] such that only reads 35 kb or longer were retained. This resulted in 77.6 Gb of reads retained, with a mean read length of 57 kb and a mean quality score of 8.1, as calculated by NanoStat v. 1.1.2[70]. These reads were then used as input for minimap2[71] v. 2.16-r922 with flag -r 10000 and subsequently miniasm[72] v. 0.3-r179. The resulting gfa file was visualized using Bandage[73] v. 0.8.1, which showed a large "hairball" of repetitive

sequences, perhaps corresponding to a relatively recent LTR (Long Terminal Repeat) retrotransposon burst that had not yet diverged enough for the assembler to tease apart (Supplementary Fig. 1). We visualized transposable elements on this graph by BLAST of a library of the genome against consensus TEs reported by RepeatModeler2[74]. Furthermore, using the EDTA repeat annotation pipeline, we found that this initial genome assembly consisted of over 77% repeats, 46% of which correspond to LTRs, primarily *Copia* at 30% of the total genome, followed by unknown LTRs at 9.5%, and *Gypsy* at 7.4% (Supplementary Data 1). There was also a substantial percentage of DNA transposons, with *Helitrons* comprising 8.4% of the assembly (Supplementary Data 1).

The resulting raw assembly was polished using Racon[75] v. 1.3.3. The length-filtered Nanopore reads were mapped to the assembly using minimap2 as input for Racon and this process was repeated for a total of three rounds, each time using the most polished Racon assembly. Next, Illumina reads were mapped to the 3x Racon polished assembly using bwa mem and the resulting bam was used as input for Pilon[76] v. 1.2.3. Like Racon polishing, this process was also repeated three times.

Next, because the resulting genome size was nearly two times greater than bioinformatic estimates for genome size, purge_haplotigs[77] v. 1.1.0 was used to check for and remove duplicated haplotigs. We used minimap2 to map Nanopore reads to the polished reference, and used SAMtools[78] view, sort, and index to process the sam file. The resulting plot from the purge_haplotigs hist function showed a clear diploid peak, hence the purged version was selected for further processing. The final BUSCO score was 96.1%.

### Scaffolding with Hi-C
Despite using an abundance of Nanopore reads, the assembly was still relatively discontiguous with 2874 contigs and an N50 around 650 kb. Hence, to create a more contiguous assembly, Hi-C reads for scaffolding were generated by Arima Genomics (Arima-HiC Kit, #A410030). The Hi-C reads were mapped for scaffolding using Juicer[79] v. 1.5.7 under default conditions, to attain the merged_nodups.txt file for input to 3D-DNA[80] v. 5.0.2. 3D-DNA parameters were adjusted to 100 kb for editing coarse resolution and 150 kb input fragments for polisher and splitter, with an editor stringency of 45. The final touches were applied by hand using Juicebox[81] v. 1.11.08. Quality was assessed using QUAST[82] v. 5.0.1, BUSCO, and by comparing assemblies using CoGe SynMap. The *Stenogyne calaminthoides* scaffolded genome assembly and annotation used for analyses in this study are available on CoGe (https://genomevolution.org/coge/GenomeInfo.pl?gid=58017). We note that three scaffolds in the genome assembly available in the NCBI database (accession JBBCBC000000000) have the following nucleotide sequence replaced with Ns due to suspected adaptor contamination (total 114 bp): HiC_scaffold_10 nt 22520170−22520201, HiC_scaffold_13 nt 29701829−29701855 and 33752502−33752528, and HiC_scaffold_23 nt 8432126−8432157.

### Repeat annotation and gene model prediction
The EDTA (Extensive de-novo TE Annotator) pipeline[83] v. 1.8.3 was used to mask repeats in the *Stenogyne calaminthoides* genome. Using the EDTA repeat annotation pipeline, we found that this initial genome assembly consisted of over 77% repeats, 46% of which correspond to LTRs, primarily *Copia* at 30% of the total genome, followed by unknown LTRs at 9.5%, and *Gypsy* at 7.4% (Supplementary Data 1). There was also a substantial percentage of DNA transposons, with *Helitrons* comprising 8.4% of the assembly (Supplementary Data 1).

For transcriptome assembly, the raw reads from each tissue were separately assembled de novo using Trinity[84] v. 2.6.6 with default parameters (*k*-mer *k* = 25). Additionally, de novo assemblies were produced using Trans-ABySS[85] v. 2.0.1 for *k*-mers 51-111 for every alternate increment resulting in 31 assemblies for every plant tissue. All

these assemblies were passed to EvidentialGene[86] v. 2017.12.21 to produce a single high-confidence transcriptome assembly for each tissue. These three final transcriptome assemblies were combined and passed once again to evigene to produce the final transcriptome assembly containing 181,789 transcripts for a BUSCO completeness score of 90.4%.

The annotation of the genome was carried out using a modular approach. The transcriptome assembly was splice-aligned against the genome assembly using PASA[87] v. 2.3.3 to produce reference ORFs. This was followed by the gene prediction step which involved a collection of hmm-based (genemark-es[88] v4.38, BRAKER[89] v. 2.1.2 with STAR aligner[90] v. 2.7.2b and AUGUSTUS[91] v3.3.2) and homology-based gene predictors (GeMoMa[92] v. 1.6.1 using model species *Arabidopsis thaliana* and *Populus trichocarpa*). The predictions from all these tools and repeats information were combined to produce a single high-confidence final gene prediction using EVidenceModeler[93] v. 1.1.1 containing 77,090 gene models with a BUSCO completeness score of 86%.

### Reference genome ploidy analysis and characterization
We used the CoGe[54] platform to analyze fractionation bias, synteny (using SynMap[55]) with *Stenogyne calaminthoides* and its relatives. For example, the *Stenogyne calaminthoides* genome was compared with several other representative core eudicot genomes using syntenic depth FractBias[34] plots. This approach maps percentages of genes retained post-polyploidy for each subgenome of a polyploid species relative to chromosomes of a reference species. We extracted the syntenic pair results in a text table, and modified this file to be used as connections for a Circos[94] plot. With both Circos, and CoGe SynMap, we identified which chromosomes were syntenic to each other. We further detailed the Circos plot by adding tracks for main repeat categories (*Gypsy*, *Copia*, and Helitron), as well as gene space. For all tracks, a genomic region density was added using the genomicDensity function from the R package circlize[95] v. 4.2.1 over windows of 1 Mb. Only scaffolds above 25 Mb in length were included in the Circos plot.

### Chloroplast genome mapping and de novo assembly of Illumina resequenced samples
Raw Illumina reads were trimmed using Trimmomatic[96] v. 0.38, in order to remove adapter contamination. The trimmed reads for each sample were individually mapped to the *Stenogyne calaminthoides* reference genome using bwa[97] mem v. 0.7.17, and each resulting bam file was filtered for a quality score of 20 using SAMtools[78] view, and sorted using SAMtools sort v. 0.1.19. Picard MarkDuplicates v. 2.7.1 (http://broadinstitute.github.io/picard/) was used to remove PCR duplicates from the mapped reads and report mapping statistics. Depth and width of mapping coverage was calculated using the BEDTools[98] v. 2.23.0 function genomeCoverageBed. Consensus sequences were called using samtools mpileup to attain plastid sequences for phylogenetics.

To check for any potential reference bias, plastomes were also assembled de novo using trimmed reads and NOVOplasty[99] v. 3.0. A seed of the *rbcL* gene of *Stenogyne microphylla* was used (Accession AF502024.1). Phylogenies were built using RAxML[100] provided by the CIPRES Science Gateway[101] with 1000 bootstrap replicates and visualized using FigTree v1.4.3 (http://tree.bio.ed.ac.uk/software/figtree/).

### De Novo Nuclear Assembly of Illumina resequenced samples
MaSuRCA[102] v. 3.2.7 was used to generate de novo assemblies based on Illumina data from each of the 45 mint taxa. Each assembly was quality checked using QUAST[82] to determine statistics, such as the N50. Each de novo assembly was used as a reference for its own reads and mapped using the same pipeline used for mapping to the *Stenogyne calaminthoides* reference assembly.

## Reference mapping and SNP calling of Illumina resequenced samples

The trimmed Illumina reads for each sample were individually mapped to the *Stenogyne calaminthoides* reference genome using the same pipeline used for the plastid mapping. Additionally, we calculated depth and width on a per chromosome basis, to detect potential differences in mapping to each subgenome. SNP calling was performed using GATK[103] v. 3.8 HaplotypeCaller in ERC mode for each sample to produce a g.vcf file for each sample. GenotypeGVCFs was then used to call joint genotypes and produce a combined file.

For filtration of called SNPs, GATK VariantFiltration was used with the following filter expression based on GATK recommendations: 'QD < 2.0 || FS > 60.0 || MQ < 50.0  ||  MQRankSum < −12.5  ||  ReadPosRankSum < −8.0 || SOR > 4.0'. This filtration was applied prior to all downstream analyses. Additionally, to avoid organellar contaminants and any spurious small fragments, only scaffolds greater than one megabase were used for all downstream analyses. Using VCFtools[104] a depth constraint of at least 5X coverage and no more than 500X coverage was applied to all datasets. This filtration also removed plastome and mitochondrial data since organellar reads map at much higher depth than 500X coverage. Depending on analyses performed, different datasets were generated comprising different sets of samples and SNP filtrations (see Supplementary Data 6). VCFtools flags –het and –singletons were used to calculate heterozygosity and singleton/doubleton content respectively. Heterozygosity and singleton statistics were calculated independently as well for each subgenome and each of the 30 syntenic chromosomes.

## BUSCO assessment and phylogenetics of genome assemblies

BUSCO[105] v. 4 was employed for each de novo assembly with the flag --limit 8 to retain up to 8 copies per BUSCO. For each BUSCO gene, all samples and their copies were combined into one fasta file using an in-house script. Next, these fasta files were aligned using TranslatorX[106]. To remove any low quality sections of the alignment, amino acid alignments were trimmed using trimAl[107], using the automated1 option to optimize trimming for downstream maximum likelihood analysis and the –backtrans option to convert back into nucleotides. Additionally, gaps accounting for greater than 20% were removed. Next, for each alignment a distance matrix was computed for each BUSCO gene using the DistanceMatrix function from the DECIPHER[108] package. Samples that deviated more than one standard deviation from the average were removed. This was important to filter out errors, such as bacterial genes found in the alignments. We then removed trees with less than 20 taxa. RAxML was used to generate phylogenies for each BUSCO gene. The resulting 2065 RAxML trees were combined into one file as input for ASTRAL-Pro 2[109]. Including only multi-copy BUSCO genes, 1290 cleaned BUSCO trees were used as input for GRAMPA[46], following removal of sequences that appeared as erroneous long branch lengths detected using TreeShrink[110]. The number of groups was set to ten in GRAMPA and all other parameters were left default. The best scoring tree was then opened in FigTree (http://tree.bio.ed.ac.uk/software/figtree/) and nodes h1 and h2 were labelled. Next, we found the best scoring trees for the other polyploidy events.

## Confirming subgenome identity

To further support the GRAMPA results and identify homeologous pairs of chromosomes, we calculated depth of coverage on a per-chromosome basis for each sample, explored DAGchainer[52] results produced by CoGe, and in parellel we ran SubPhaser[51] with the -intact_ltr option.

## SNP phylogenetic analyses

Phylogenetic analyses were performed on the datasets DS4 and DS4a-c (Supplementary Data 6). A maximum likelihood tree was generated using RAxML[100] v. 8.0.0 including adjustments for ascertainment bias (--asc-corr lewis) as recommended for SNP data[111] and 500 bootstrap replicates. Trees were visualized and edited using FigTree (http://tree.bio.ed.ac.uk/software/figtree/). Additionally, SplitsTree4[112] v. 4.16.2 was used to generate a NeighborNet[57,113] network of datasets DS4 and DS4b (Supplementary Data 6) using LogDet[114] distances.

Using the R package ape[115], newick trees for the BUSCO and SNP phylogeny based on the full dataset (DS4) were converted into ultrametric trees using the function chronos. The dendextend R package[116] was used to create tanglegrams. The untangle function was used to find the best untangling method as reported with the entanglement function score. The tanglegram function was used to generate the final figures.

Furthermore, using a subset of Hawaiian mints and their closest relatives, phylogenies were generated based upon chromosomes and summarized using the DensiTree[58] function implemented in R package phangorn[117]. Additionally we generated non-overlapping windows of 25,000 SNPs, resulting in 287 trees, also summarized with DensiTree.

## Twisst

*Twisst*[56] (topology weighting by iterative sampling of subtrees) is a method that estimates phylogenies based on sliding windows of SNPs along the genome and quantifies the contribution of a given topology to a full tree via weighting. SNP trees were prepared as recommended by the *Twisst* software and included the phasing step using beagle[118] v. 4.1. In the *Twisst* program, the maximum number of monophyletic groups is eight so that the number of potential trees is computationally feasible. Hence, we created two datasets to address different questions. First, we used all samples (dataset DS10), assigning a monophyletic group to *Stachys byzantina, S. coccinea, S. sylvatica*, ENAA *Stachys*, WNA *Stachys*, and one group for each Hawaiian mint genus. Second, we limited our analysis to the Hawaiian mints and their WNA relatives (dataset DS10a), which all have the same ploidy level. Here, we made subsets of the data into seven monophyletic clades based on the SNP tree. WNA *Stachys* made up one group, *Phyllostegia* was separated into two monophyletic groups and *Stenogyne* into three monophyletic groups based on the SNP phylogeny (partitions shown in Supplementary Fig. 15). We used 50 SNP non-overlapping windows as recommended[56]. We used the R script provided with *Twisst* to plot the results by chromosome and printed the weight below each tree.

## TreeMix and ADMIXTURE analyses

We ran TreeMix[63] on DS4c (*Stachys byzantina* excluded; Supplementary Data 6). TreeMix was run for m values 0 to 10, in each case with the --noss option included as this was run on an individual level and not a population level.

ADMIXTURE[64] v. 1.3 for K values 2–10 were used along with the –cv option to find the optimal (lowest cross validation) K value for the number of ancestral groups for datasets DS4, DS4a, and DS4b, as well as data set HM1 (Supplementary Data 6). The ADMIXTURE results were plotted using the barplot function in R, and the cv values were plotted in ggplot2 (https://github.com/tidyverse/ggplot2).

## Principal component analysis (PCA)

PCA was run under both an individual and genus level for datasets DS4, DS4b, and DS4c, as well as dataset HM1 (Supplementary Data 6). The eigensoft package[59] v. 6.1.3 convertf was used to convert plink.map and.ped files into.ind,.geno, and.snp files. Then, the smartpca.perl script was used to run PCA for PC1 to PC10 under default parameters. To plot the results, the ggplot2 function geom_point() was used along with the R package ggrepel[119] v. 3.5.1.

## $f_3$ statistics

The $f_3$ statistics were calculated using the admixtools[62] package with datasets DS7 (all 45 samples of Hawaiian mints and relatives) and HM4 (all 127 *Stenogyne* samples) as input with missing data allowed. First,

**Article**

using the vcftools --plink option, the vcf file was converted into ped and map files as input for convertf. Every possible combination of three samples was tested. The $f_3$ Z-scores were converted into p-values for correction using the p.adjust function and an FDR method[53,120] and converted back into Z-scores. We used geom_tile function in the R package ggplot2 (https://github.com/tidyverse/ggplot2) to plot the corrected Z-scores.

## Morphometric analyses

The following morphological traits for all samples of *Stenogyne microphylla* and *S. rugosa* and their purported hybrids on Mauna Kea: trichome length, stem width, average leaf area, and average width and length ratios across three leaves for each sample (Supplementary Data 12). Measurements were recorded using Nikon SMZ25, a motorized multi-focus stereo microscope with a ring fiber illumination set (C-FLED2 LED light source) attached, at the Singapore Botanic Gardens. Leaf area, length, and width were calculated. Per each sample, three leaves were used at random for measurements. Additional measurements were taken, such as stem width. To test for potential correlation among morphological traits and principal components based on SNP data, principal components from SNPs were added to the measurements file and used to search for correlations using the pairs.panels function from the R package psych. Furthermore, principal components from SNPs and morphological data were input into PAST[121] v. 4.11 to create scatterplots. Correlation was tested using the Pearson correlation coefficient (r).

## Reporting summary

Further information on research design is available in the Nature Portfolio Reporting Summary linked to this article.

## Data availability

The genome data generated in this study have been deposited in the NCBI database under BioProject accession code PRJNA924716 and BioSample ID SAMN32782865. For the reference genome of *Stenogyne calaminthoides*, Hi-C reads are under accession SRR23341345, RNA-seq data under accession SRR23341344, Illumina shotgun data under accession SRR23341343, and Oxford Nanopore reads under accession SRR23341342. This Whole Genome Shotgun project has been deposited at GenBank under the accession JBBCBC000000000. The *Stenogyne calaminthoides* genome assembly and annotation used for analyses in this study is available on CoGe [https://genomevolution.org/coge/GenomeInfo.pl?gid=58017]. Raw reads for resequenced samples can be found under accession numbers SAMN32767766-SAMN32767919. Additional processed data are available in Dryad [https://doi.org/10.5061/dryad.ghx3ffbwc]. Processed data generated in this study and used for main text figures are provided in source data files. Source data are provided with this paper.

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

## Acknowledgements

The authors acknowledge financial support from the School of Biological Sciences of Nanyang Technological University, U.S. National Science Foundation awards 2139311 to C.L. and 2030871 to V.A.A. and Research Council of Norway award 154145 to V.A.A. The authors thank Singapore Botanic Gardens for laboratory support, Lyman Perry for permission to collect on Hawaiian State lands, and Patty Moriasu at the Volcano Rare Plant Facility, the National Tropical Botanical Garden, Elizabeth Stacy and Jennifer Johansen for assistance with plant specimens. We dedicate this paper to the memory of Timothy J. Motley, a great friend, colleague and specialist on the Hawaiian flora.

## Author contributions

V.A.A. and C.L. designed research; C.M.T., S.R., J.T.S, Y.W.L., J.S., T.P.M., V.A.A. and C.L. performed research or analyzed data; A.-C.S., M.B., Y.W.L., J.S. and T.P.M. contributed reagents, materials, and/or tools; C.M.T., V.A.A. and C.L. wrote the paper.

## Competing interests

The authors declare no competing interests.
