## [Peer Review File · Nature Communications]

Allopolyploid origin and diversification of the Hawaiian endemic mintsReviewers' Comments:

Reviewer #1:

Remarks to the Author:

The Hawaiian endemic mints provide an important context for studying lineage migration, speciation, and extinction on isolated islands. This paper evaluates a comprehensive study that focuses on the assembly of a high-quality reference genome for *Stenogyne calaminthoides*, a Hawaiian mint species, using DNA reads obtained from Oxford Nanopore and Illumina sequencing technologies, in combination with HiC-scaffolding. The researchers also conducted resequencing for 30 other Hawaiian mint taxa and 15 related species in the *Stachys* genus. Notably, they explored a putative hybrid swarm of *Stenogyne rugosa* and *S. microphylla*, consisting of 109 individuals found on Mauna Kea²⁴, and documented their morphological traits. By employing these genomic resources, the study delved into the evolutionary history of the Hawaiian mint lineage, examining genomic signatures and the influence of polyploidization, taxonomic radiation, and hybridization events from ancient to recent times. The findings shed light on the fascinating evolutionary dynamics within this plant group. However, I have following major comments that needs to be address prior to the acceptance of the manuscript.

1. The evidence for a recent WGD event is not fully convincing based on the weak cross-linking between some chromosomes in the Hi-C plot and high similarity between chromosomes in the Circos plot. These patterns could reflect haplotype redundancy rather than a WGD event. The authors should provide the original k-mer frequency plot and results from genome scope or other tools to allow manual inspection of the peaks for polyploid genomes.
2. Even after genome deduplication, the current assembly may still represent a duplicated haplotype rather than a true haploid. The `purge_haplotigs` software is mainly for diploids, so parameters like `-m` need adjustment for suspected polyploids.
3. The genome may be an allotetraploid (AABB) with two more similar subgenomes (AB) and two more divergent ones (A1B1 versus A2B2). It seems the authors have assumed two of the haplotypes (AB), since the two haplotypes are not very similar (as opposed to the other two haplotypes). A1 vs B1 compare with A1 vs A2). The authors need to provide updated results to support the WGD claim. For example, try fluorescence microscopy, try manually counting chromosomes to determine the true chromosome number and ploidy of the species. Authors need to be very careful with this result, as it relates to the most important results and conclusions of the entire article.
4. For phylogenetics, SNP conflicts likely reflect haplotype versus species differences. The authors should filter SNP-rich regions and use more conserved loci to construct trees. Many conflicts are between close species, suggesting haplotype effects.
5. The authors used different methods to evaluate the quality of the assembled genomes and obtained good results. However, they do not use the LTR Assembly Index (LAI), an important criterion for the genome quality nowadays <https://academic.oup.com/nar/article/46/21/e126/5068908>.
6. Overall, further analyses are needed to confirm ploidy, distinguish haplotype from species differences, and strengthen support for a WGD event in this lineage. More information would clarify the relationships between Hawaiian species and the roles of polyploidy and hybridization in their diversification.

Reviewer #2:

Remarks to the Author:

This paper investigates the evolutionary history of an exemplar island plant radiation in the Hawaiian

mints and their continental relatives. The authors reconstruct the first genome for a species of *Stenogyne* to elucidate the history of polyploidy in this clade. They further use resequencing (genome skimming) via Illumina data for most species in the Hawaiian mint clade along with representatives of *Stachys* from continental North America. These data were used to reconstruct SNP-based phylogenies to examine species relationships and the influence of hybridization and polyploidy among the species. A final focus was to examine the level of admixture (hybridization) between two species that occur along a transect on one island (Mauna Kea). Overall, the amount of data generated is impressive and I think the genome of *Stenogyne* alone and its analysis would make for a solid publication. It would be helpful to the reader if the three main goals/aims of the study are clearly explained in the introduction and throughout, the links to the various datasets are made clear.

In the introduction, several questions are posed (lines 78-79) related to speciation but these are not answered, so it's not clear what the relevance is to the present study. Instead the focus is on hybridization and allopolyploidy within the Hawaiian mint clade, which has been the focus of previous phylogenetic study. At the end of the introduction, it would be helpful to the reader if the authors could explicitly outline the questions being addressed by the many sources of data that are generated in the study. Moreover, throughout the results/discussion it is difficult to know what data set is being referred to and the presentation of the methods does not lead the reader back to specific questions. For example, RNAseq was done on separate tissues of *Stenogyne calaminthoides* (lines 656-658), the species for which a whole genome was reconstructed, but it is not clear that these data were used or how (or why!).

Throughout the use of 'S.' is often confusing as it is not clear whether this refers to *Stachys* or *Stenogyne*. It would be helpful to the readers, who don't have the species memorized, to spell the genus names in full.

Line 94 – suggest adding 'Hawaiian' to the start of the sentence regarding plant radiations. Other island plant radiations have similarly been studied in the Canary Islands and New Zealand for example (which also exhibit hybridization and polyploidy).

Lines 101-108 – descriptions of the genera should be parallel, reflecting habit, flower color, pollinators, etc. It would also be interesting to know their geographic distributions – are they endemic to particular islands?

Line 108 – remove comma after prior and suggest removing 'marker sparse'.

Line 122 – it would be nice to know more about the 'morphological radiation' of this group – perhaps additional details could be added to lines 101-108 so that these features are appreciated by the reader.

Lines 117, 222, 403 - allopolyploid hybrid is redundant – suggest changing these to allopolyploid

Line 296 – *Stschys* should be *Stachys*

Line 540 – this species was not sampled as part of the study – is that why it is referred to as a 'ghost' taxon?

Line 548 – suggest using an appropriate botanical term to replace 'fuzzy'

Line 619 – morphologically should be morphological

Line 660 – silic should be silica

Line 764 – suggest changing chloroplasts to plastomes

Line 802 – think a verb is needed here after computer (or instead of)

Line 885 – include full genus name

For genome assembly, were organellar data (plastome and mitochondria) extracted prior to assembly? Chloroplast genome reconstruction was mentioned in the methods (lines 754-768), but it was not clear if organellar data were extracted from samples for other analyses (i.e., SNPs).

Figure 2 – Spell out genus names or label genera in some other way – *Stenogyne* and *Stachys* are both represented by 'S.' No support values are shown on the SNP tree. Is this the RAxML tree? It would be helpful to label the ADMIXTURE plots at the top (as the SNP and BUSCO are labelled). It is really difficult to figure out where the tanglegram is – I'm used to seeing these at the tips and given the small size of the figure, the dashed lines are hard to see. I wonder if a small inset could be shown to highlight these? Or are the ADMIXTURE crossed bars showing this? If so, an explanation in the legend would be helpful. What do the numbers in parentheses mean after some taxa?

Figure 3 – what data are shown here? "this dataset"

Figure 5 b – helpful to indicate in the legend that the numbers along the side refer to the map and transect in a.

Response to Referees

We thank the reviewers for their insightful comments, and the editor for the opportunity to resubmit a modified version of our manuscript. We feel we have been able to successfully address all points in our responses below.

Sincerely, for all authors, Charlotte Lindqvist

Reviewer #1 (Remarks to the Author):

The Hawaiian endemic mints provide an important context for studying lineage migration, speciation, and extinction on isolated islands. This paper evaluates a comprehensive study that focuses on the assembly of a high-quality reference genome for *Stenogyne calaminthoides*, a Hawaiian mint species, using DNA reads obtained from Oxford Nanopore and Illumina sequencing technologies, in combination with HiC-scaffolding. The researchers also conducted resequencing for 30 other Hawaiian mint taxa and 15 related species in the *Stachys* genus. Notably, they explored a putative hybrid swarm of *Stenogyne rugosa* and *S. microphylla*, consisting of 109 individuals found on Mauna Kea²⁴, and documented their morphological traits. By employing these genomic resources, the study delved into the evolutionary history of the Hawaiian mint lineage, examining genomic signatures and the influence of polyploidization, taxonomic radiation, and hybridization events from ancient to recent times. The findings shed light on the fascinating evolutionary dynamics within this plant group. However, I have following major comments that needs to be address prior to the acceptance of the manuscript.

1. The evidence for a recent WGD event is not fully convincing based on the weak cross-linking between some chromosomes in the Hi-C plot and high similarity between chromosomes in the Circos plot. These patterns could reflect haplotype redundancy rather than a WGD event. The authors should provide the original k-mer frequency plot and results from genome scope or other tools to allow manual inspection of the peaks for polyploid genomes.

Thank you for your particular interest in establishing polyploid status for our reference assembly. In fact, we included multiple lines of evidence already in our first submission that support the conclusion that our *Stenogyne calaminthoides* chromosome-scale genome displays underlying polyploid structure rather than being a diploid assembly. Here, we review these for the sake of clarity.

First, we draw the reviewer's attention to our results from the SubPhaser application, which we successfully used to phase the two subgenomes of the recent-most polyploidy event we detected. SubPhaser uses a kmer-based approach, as does GenomeScope, which the reviewer recommends. Indeed, SubPhaser results were presented in our main text Fig. 3B, and in Supplementary Fig. 13. The main text figure panel is reproduced here for clarity:

Response Fig. 1. Panel B of manuscript Fig. 3 shows successful phasing of the recent-most allopolyploidization event in *Stenogyne calaminthoides*. Note the strong differences in clustering behavior for the yellow versus blue subgenomes, which no doubt reflects different LTR transposon compositions that could not be haplotypic.

The dendrogram to the left, which completely separates two subgenomes, reflects the PCA analysis to the right, which is based on kmer distributions that result from substantially different LTR retrotransposon compositions (Response Fig. 1). It is therefore clear from these results that the yellow and blue subgenomes are not only distinctly phased, but that they also possess strongly different LTR compositions, which also helps to validate our contention made in the manuscript that the recent-most whole genome duplication event was an allopolyploidization. The underlying LTR structures between the yellow and blue subgenomes are not only broadly separated on PC1, but the blue subgenome is much more scattered along PC2, likely reflecting more variance in kmer composition in that subgenome as compared to that of the yellow subgenome. These SubPhaser results alone provide great confidence in the polyploid (and indeed the allopolyploid) nature of our reference assembly.

A further demonstration of *Stenogyne's* recent-most whole genome duplication is apparent from our use of the FractBias tool that accompanies the SynMap syntenic dotplot application in the CoGe platform. In main text Fig. 1D (see also Supplementary Fig. 2), it can be seen that each of the four *Stenogyne* chromosomes that map against *Vitis vinifera* has different fractionation (i.e., alternate gene loss since polyploidization) organization, yielding firm evidence for the assembly's paleo-octaploid structure. We reproduce this panel below for clarity (Response Fig. 2):

Response Fig. 2. Panel D of manuscript Fig. 1 demonstrates that each of the 4 *Stenogyne* scaffolds that map to this particular *Vitis* chromosome has accrued different fractionation patterns since the 2 polyploidy events specific to the *Stenogyne* lineage. See response text below for further details.

Here, it can be seen that the two gray-scale scaffolds, and the two blue-scale scaffolds, have different fractionation patterns, both relative to each other as 1:1 pairs, as well as between the pairs via all-gray vs. all-blue. The gray-scale and blue-scale pairs each represent derivatives of the recent-most polyploidy event within *Stenogyne*, while gray together vs. blue together represent the early, core-Lamiales polyploidy event we describe in the main text. The fractionation differences between gray/blue scaffolds stemming from this earlier event are much more substantial than within-gray or within-blue fractionation distinctions, but the latter are still far more extensive than could possibly occur between haplotypes in a diploid assembly. Had the gray and blue pairs been haplotypes instead of subgenomes, the plant individual sampled would have had strongly uneven gene content in its haploid halves, and thus could not have been viable.

Further validating our conclusions of polyploid status for the reference genome are our kmer-based GenomeScope and Smudgeplot analyses, which we chose not to include for space conservation reasons as they are redundant with the SubPhaser kmer approach. Nonetheless, we report them here for clarity. GenomeScope shows two very distinct 21-mer distributions, comparable to what we observed using SubPhaser (Response Fig. 3). Smudgeplot clearly predicted paleo-octoploid structure, as we inferred using multiple other approaches and reported in the paper (Response Fig. 4).

Response Fig. 3. GenomeScope kmer profile for our reference assembly shows two distinct peaks, which is fundamentally comparable to our SubPhaser results.

Response Fig. 4. Smudgeplot results, also stemming from kmer analysis, clearly predicts paleo-octoploid status for our reference assembly.

2. Even after genome deduplication, the current assembly may still represent a duplicated haplotype rather than a true haploid. The `purge_haplotigs` software is mainly for diploids, so parameters like `-m` need adjustment for suspected polyploids.

We thank the reviewer for suggesting a closer look at how the use of the Purge Haplotigs software may be affected by its application to polyploid genomes. Some of the present authors recently published the genome of the carnivorous pitcher plant *Nepenthes gracilis* (see Nature Plants publication: <https://www.nature.com/articles/s41477-023-01562-2>) – there, we successfully used the Purge Haplotigs application with default parameter settings, i.e., including the default `-m` value, to filter diploid contigs from our primary Oxford Nanopore based assembly to yield the expected haploid genome size for the species. *Nepenthes gracilis* has a highly complex polyploid history, possessing 5 subgenomes and therefore decaploid structure in its diploid state; as such, Purge Haplotigs can be highly successful even for far more complex polyploids than *Stenogyne*, even if the software wasn't specifically designed to be used with them. In yet another paper of ours on the eucalyptus relative *Syzygium grande*, which has paleotetraploid structure, we similarly used Purge Haplotigs with default settings to filter diploid contigs from our primary Oxford Nanopore based assembly (see Nature Communications publication: <https://www.nature.com/articles/s41467-022-32637-x>). Moreover, the lengths of our pre- and post-purging *Stenogyne* assemblies presented in the current manuscript closely enough approximate 2:1 at 2,450,994,252 to 1,415,487,396 bases, respectively. Because the “diploid” assembly is almost 450 Mb smaller than twice the haploid assembly, it is reasonable to suppose that the former was only partially diploid, having some contigs collapsed to haploid status.

We therefore feel confident that our use of Purge Haplotigs deleted predominantly diploid segments of our primary assembly, revealing instead the polyploid structure of the *Stenogyne* genome as we reported in the first submission.

Nonetheless, we performed an additional check to further underscore this point by comparing synonymous substitution (Ks) values for syntenic gene pairs between the pre-purged (diploid) *Stenogyne* assembly against *Vitis vinifera* in comparison with a corresponding histogram using our haploid reference assembly against *Vitis*. For this purpose, we predicted gene models on the diploid assembly using the proteome of our haploid reference. As can be seen in the Response Fig. 5 below, there is a clear haplotypic split (allelic instead of duplicate gene pairs) at very low Ks value in the histogram resulting from syntenic analysis of the pre-purging diploid assembly against *Vitis*. Other than the low Ks purple mode (see caption of Response Fig. 5), this Ks histogram is comparable to the similarly-scaled histogram stemming from use of the *Stenogyne* haploid reference assembly. This experiment further validates our appropriate use of Purge Haplotigs, and therefore our interpretation of syntenic results from our *Stenogyne* reference assembly as representing polyploid structure (please refer to additional description of these results in the caption of Response Fig. 5). While we present this check here for completeness, we feel it is redundant to an already lengthy paper given other strong points already made in the text.

A

B

Response Figure 5. Histograms of syntenic gene pairs between *Stenogyne calaminthoides* and *Vitis vinifera*, as calculated using CoGe's SynMap tool. Coloration is by synonymous substitution (Ks) values. Ks values, as is customary, can be interpreted as time since syntenic gene pair split, with low values being the most recent. **A**, our *Stenogyne* reference genome (1,415,487,396 bases, contigs only) against *Vitis*; **B**, our pre-purging diploid assembly (2,450,994,252 bases, scaffolded with HiC) against *Vitis*, the gene models for which have been repredicted using GeMoMa using the reference proteome as reprediction input. The scales of both histograms are restricted to $\log_{10} 1, -1$ on the x-axis for clarity. In the haploid assembly, **A**, the peak with the green mode and right skew represents the *Stenogyne-Vitis* species split (the mode itself) and underlying ancestral polyploidy events partly obscured by this orthologous synteny (the right skew). In the diploid assembly, **B**, the green mode similarly shows the species split, very close in Ks value to the green haploid mode in **A**. The slight modal differences and differences in numbers of syntenic pairs discovered little doubt stem from having used the haploid reference proteome to predict diploid gene models, imperfectly so, and due to the result that two haplotypes contribute to the syntenic matches with *Vitis* in **B**. In **B**, the blue mode at very low $\log_{10} Ks$ represents syntenic gene pairs aligning from haplotypic contigs – i.e., it represents the recent orthologous (allelic) split of the two haploid halves of the diploid *Stenogyne* assembly. The sharp purple mode most likely represents irrational Ks values stemming from imperfectly repredicted haplotypic *Stenogyne* gene models poorly aligning against their *Vitis* counterparts. Haplotypic splits of this type in diploid assemblies are commonly seen in incompletely purged assemblies, and hence the requirement of careful purging to haploid status to generate reference quality genomes.

3. The genome may be an allotetraploid (AABB) with two more similar subgenomes (AB) and two more divergent ones (A1B1 versus A2B2). It seems the authors have

assumed two of the haplotypes (AB), since the two haplotypes are not very similar (as opposed to the other two haplotypes). A1 vs B1 compare with A1 vs A2). The authors need to provide updated results to support the WGD claim. For example, try fluorescence microscopy, try manually counting chromosomes to determine the true chromosome number and ploidy of the species. Authors need to be very careful with this result, as it relates to the most important results and conclusions of the entire article.

As responded to above, we were able to establish allopolyploidization underlying the recent-most whole genome duplication event using SubPhaser and FractBias alone. Concerning confirmatory chromosome counts, these are indeed available and were in fact reported in the main text submission version, line 174: “After scaffolding with Hi-C reads (Fig. 1b), the final assembly had an N50 over 37 Mb and contained 434 scaffolds, including 32 scaffolds larger than 25 Mb, closely matching the expected chromosome count of $n = 32$ ”. This chromosome number was reported in the following reference we provided:

Carr, G.D. Chromosome evolution and speciation in Hawaiian flowering plants. *Evolution and speciation of island plants*, 5-47 (1998).

4. For phylogenetics, SNP conflicts likely reflect haplotype versus species differences. The authors should filter SNP-rich regions and use more conserved loci to construct trees. Many conflicts are between close species, suggesting haplotype effects.

Firstly, we already included in our first submission a coalescent species tree based on genes recovered from BUSCO search. BUSCO genes, used to check gene space prediction quality, are indeed strongly conserved loci anticipated to be single-copy across species. In fact, our BUSCO species tree is compared directly with our SNP-based phylogenetic result in main text Fig. 2, which we reproduce below for clarity (Response Fig. 6). The “tanglegram” comparison in between these trees (which also highlights the results of our ADMIXTURE analysis) demonstrates strong congruence between the (genome-wide average) SNP-based maximum likelihood phylogenetic approach and coalescent-based species tree methodology used to analyze BUSCO genes. Further, but considered redundant for reporting in the paper, we filtered our SNPs for homozygous SNPs only, phylogenetic analysis of which yielded an identical tree topology (results not shown). In summary, we are confident that our SNP data are well representative of the genome as a whole, for phylogenetic reconstruction as well as other inferences presented in the manuscript.

Response Figure 6. Main text Fig. 2 shows strong congruence between phylogenetic trees based on SNPs (left) and highly conserved BUSCO genes (right).

NOTE: Response Fig. 5 depicts the submitted version of main text Fig. 2; in response to comments by Reviewer 2, below, we have updated the graphic – see that also below.

5. The authors used different methods to evaluate the quality of the assembled genomes and obtained good results. However, they do not use the LTR Assembly Index (LAI), an important criterion for the genome quality nowadays <https://academic.oup.com/nar/article/46/21/e126/5068908>.

We thank the reviewer for this suggestion. LAI and SubPhaser, already reported on above, both use LTRharvest. We used the LTRharvest results from our SubPhaser analysis and proceeded to run LTRfinder (with LTRharvest and LTRretriever and RepeatModeler as input) for LAI.

Results from the entire HiC-scaffolded assembly are as follows:

Intact	0.0014
Total	0.5021
raw_LAI	0.28
LAI	2.31

Here, the raw LAI is (complete LTR length/all LTR length) * 100

LAI is raw LAI + 2.8138 × (94 – whole genome LTR identity)

This LAI is not as high as anticipated for a high-quality assembly, but polyploid assemblies should always be investigated subgenome-wise for LAI (as discussed here

for autopolyploids: https://github.com/oushujun/LTR_retriever/issues/145 - and the same approach is obviously extendable to suspected allopolyploids as well). Thus, we endeavored to check values for our two well-phased subgenomes independently.

Subgenome-wise, the LAIs increased drastically into the gold-standard range for individual subgenomes:

ENA subgenome

Intact	0.0633
Total	0.0739
raw_LAI	85.66
LAI	87.17

UC subgenome

Intact	0.0609
Total	0.1119
raw_LAI	54.37
LAI	54.53

In the interest of space conservation, we chose not to include these confirmatory results regarding high genome quality in our manuscript.

6. Overall, further analyses are needed to confirm ploidy, distinguish haplotype from species differences, and strengthen support for a WGD event in this lineage. More information would clarify the relationships between Hawaiian species and the roles of polyploidy and hybridization in their diversification.

Please find several confirmatory responses above that we trust the reviewer will find convincing.

Reviewer #2 (Remarks to the Author):

This paper investigates the evolutionary history of an exemplar island plant radiation in the Hawaiian mints and their continental relatives. The authors reconstruct the first genome for a species of *Stenogyne* to elucidate the history of polyploidy in this clade. They further use resequencing (genome skimming) via Illumina data for most species in the Hawaiian mint clade along with representatives of *Stachys* from continental North America. These data were used to reconstruct SNP-based phylogenies to examine species relationships and the influence of hybridization and polyploidy among the species. A final focus was to examine the level of admixture (hybridization) between two species that occur along a transect on one island (Mauna Kea). Overall, the amount of data generated is impressive and I think the genome of *Stenogyne* alone and its analysis would make for a solid publication. It would be helpful to the reader if the three main goals/aims of the study are clearly explained in the introduction and throughout, the links to the various datasets are made clear.

We thank the reviewer for suggesting that we explicitly spell out the goals of our work. We have now added a paragraph to that effect at the end of the Introduction, as follows:

Lines 134-141: “In this study, we had three major goals: to (i) generate a high-quality, chromosome-level genome of *Stenogyne calaminthoides* (**Fig. 1a**) to investigate polyploid history of the Hawaiian mint lineage, (ii) use this reference genome and resequencing of multiple species to establish the origin and phylogeny of the Hawaiian mints and their mainland relatives, as well as potential admixture history, and (iii) investigate recent introgression among *Stenogyne* species that co-occur at high elevation on Mauna Kea²⁴. Using these genomic resources, we evaluate whether geographic speciation, a drift-biased process that may limit adaptive interpretations, played a major role in the Hawaiian mint radiation.”

In the introduction, several questions are posed (lines 78-79) related to speciation but these are not answered, so it's not clear what the relevance is to the present study.

Perhaps our explicit discussion of evidence for geographic speciation was missed by the reviewer. For examples, please refer here (“speciation” in **bold** text):

Lines 437-441: “To gain resolution for ingroup taxa, we subsequently removed *Stachys byzantina* and found that a rough cline appears shared by the rest of *Stachys* and the Hawaiian mints, one that is especially linear for Hawaiian mints and their close WNA relatives (**Fig. 4a**, Supplementary Fig. 19b). Such clinal variation may correspond to progressive cladogenesis via **geographic speciation**²².”

And here:

Lines 456-461: “The clinal variation observed among the Hawaiian mints may correspond to progressive cladogenesis via **geographic speciation**²². For example, allelic variation may have become fixed along an ongoing cladogenetic process caused by serial founder events in an island-hopping model of **geographic speciation**⁵⁶. Such

relatively simple isolation-by-distance processes may be facilitated in an extremely young and rapidly expanding and dissecting volcanic environment, such as that of the Hawaiian Islands.”

And in our paper’s summarizing paragraph:

Lines 630-636: “In summary, our work is consistent with a model of **parapatric speciation** associated with founder events, concomitant with rapid environmental changes in a dynamic volcanic landscape. The allopolyploid ancestry of the Hawaiian radiation may have set up an extensive underlying genomic diversity that could have fueled morphological distinctions driven by drift alone. However, such rapid evolutionary radiations can set up a “nightmare scenario” for disentangling phylogenetic discordances caused by ILS, wherein allelic inheritance may not follow the cladogenetic sequence of events.”

Instead the focus is on hybridization and allopolyploidy within the Hawaiian mint clade, which has been the focus of previous phylogenetic study. At the end of the introduction, it would be helpful to the reader if the authors could explicitly outline the questions being addressed by the many sources of data that are generated in the study.

Please see above with regard to our new final paragraph of the Introduction.

Moreover, throughout the results/discussion it is difficult to know what data set is being referred to and the presentation of the methods does not lead the reader back to specific questions.

Thank you for suggesting that we clarify data set presentation.

Supplementary Table 6 provides a new summary that outlines the different datasets and their use among the various analyses performed. This table now clarifies which datasets were used for analyses and associated figures as presented in the main text and the supplementary material. Methods text and figure captions have also been changed to reflect which datasets were used.

For example, RNAseq was done on separate tissues of *Stenogyne calaminthoides* (lines 656-658), the species for which a whole genome was reconstructed, but it is not clear that these data were used or how (or why!).

RNAseq from different tissues was not used for expression profiling, but rather to generate a transcriptome for gene annotation purposes. Different sets of transcripts are expected from roots, stems, and young leaf tissue, and the greatest global completeness as possible is preferable for gene space discovery.

Throughout the use of ‘S.’ is often confusing as it is not clear whether this refers to *Stachys* or *Stenogyne*. It would be helpful to the readers, who don’t have the species memorized, to spell the genus names in full.

We appreciate the reviewer pointing out the potential confusion readers may have with the Use of *S.* for both *Stenogyne* and *Stachys*. We have now replaced almost all listings

of “S.” with full genus name unless they clearly occur as a species list following the full genus name.

Line 94 – suggest adding ‘Hawaiian’ to the start of the sentence regarding plant radiations. Other island plant radiations have similarly been studied in the Canary Islands and New Zealand for example (which also exhibit hybridization and polyploidy).

Thank you; “Hawaiian” has now been inserted.

Lines 101-108 – descriptions of the genera should be parallel, reflecting habit, flower color, pollinators, etc. It would also be interesting to know their geographic distributions – are they endemic to particular islands?

Thank you; we have now made certain that parallel descriptions – fruit, floral, and habitat – are provided. We also added a sentence that most individual species are confined to just a single island in the archipelago. We do not review biogeography of the group as a whole here, but instead provide specifics only for the species actually analyzed (in Supplementary Table 4 and Supplementary Fig. 8).

Line 108 – remove comma after prior and suggest removing ‘marker sparse’.

We felt that the point was important to make, but we have now reworded the sentence into better English:

Lines 117-119: “Prior phylogenetic analyses of the Hawaiian mint lineage, based on single or few loci, discovered that the Hawaiian endemic mints form a monophyletic group nested within the nearly global genus *Stachys*.”

Line 122 – it would be nice to know more about the ‘morphological radiation’ of this group – perhaps additional details could be added to lines 101-108 so that these features are appreciated by the reader.

Again, our intent has not been to provide comprehensive information on the lineage; however, we have added more description in this paragraph to better inform readers:

Lines 101-116: “Among the most species-rich angiosperm radiations to occur on the Hawaiian Islands is that of the Hawaiian endemic mint lineage (Lamiaceae), which exhibit considerable diversity in fruit, floral, and habitat features leading to their classification of ~60 species assigned to the three genera, *Haplostachys*, *Phyllostegia*, and *Stenogyne*^{23,24}. *Haplostachys*, which consists of five species with only a single extant taxon, *H. haplostachya*, is the only genus with dry fruits; the other genera bear fleshy fruits, which are rare in Lamiaceae. *Haplostachys* and *Phyllostegia* (34 species) both have mostly pink to white, fragrant flowers with a prominent lower corolla lip, typically associated with insect pollination. *Stenogyne* (21 species) is unique in that it has primarily pink to red nectar-producing flowers with a reduced lower lip and a longer corolla tube, typical of bird pollination. The distribution of *Haplostachys* has today been reduced to relatively small, restricted subpopulations in the xerophytic shrubland at low-mid elevation between Mauna Loa and Mauna Kea on Hawai’i²⁵. *Phyllostegia* and *Stenogyne* are widely distributed on all the extant high islands of Hawaii, primarily occurring in wet/mesic forest environments, although a few species of *Stenogyne* are

found in subalpine zones of Haleakala, Mauna Kea, and Mauna Loa²⁴. Most individual species are confined to just a single island in the archipelago.”

Lines 117, 222, 403 - allopolyploid hybrid is redundant – suggest changing these to allopolyploid

Thank you; done.

Line 296 – Stschys should be Stachys

Thank you; done.

Line 540 – this species was not sampled as part of the study – is that why it is referred to as a ‘ghost’ taxon?

Thank you; we replaced “ghost” with “unsampled” throughout.

Line 548 – suggest using an appropriate botanical term to replace ‘fuzzy’

Thank you; we replaced “fuzzy” with the proper botanical term “tomentose”.

Line 619 – morphologically should be morphological

Thank you; corrected.

Line 660 – silic should be silica

Thank you; corrected.

Line 764 – suggest changing chloroplasts to plastomes

Thank you; corrected.

Line 802 – think a verb is needed here after computer (or instead of)

Thank you; we have altered the sentence as follows: “Given the large number of Nanopore raw reads and our computational limitations, the reads were first filtered using NanoFilt⁶⁷ such that only reads 35 Kb or longer were retained.”

Line 885 – include full genus name

Thank you; done.

For genome assembly, were organellar data (plastome and mitochondria) extracted prior to assembly? Chloroplast genome reconstruction was mentioned in the methods (lines 754-768), but it was not clear if organellar data were extracted from samples for other analyses (i.e., SNPs).

Thank you; we have added a clarifying sentence as follows:

Lines 800-802: “Using VCFtools¹⁰¹ a depth constraint of at least 5X coverage and no more than 500X coverage was applied to all datasets. This filtration also removed

plastome and mitochondrial data since organellar reads map at much higher depth than 500X coverage.”

Figure 2 – Spell out genus names or label genera in some other way – *Stenogyne* and *Stachys* are both represented by ‘S.’

Thanks, generic names have been edited to the following: “S.” is now substituted with “*Sta.*” for *Stachys*, “*Ste.*” for *Stenogyne*. Moreover, “*Phy.*” is now substituted for *Phyllostegia*, and “*Hap.*” for *Haplostachys*. This is described in the figure caption.

No support values are shown on the SNP tree. Is this the RAxML tree?

All nodes in the SNP tree, except the one supporting monophyly of *Phyllostegia*, are supported with a bootstrap of 100%. This is shown in Supplementary Fig. 7. Indeed, the SNP tree is a RAxML tree and this information has been added to the figure caption.

It would be helpful to label the ADMIXTURE plots at the top (as the SNP and BUSCO are labelled).

Thank you; the ADMIXTURE plot in the tanglegram is now labeled “ADMIXTURE” (Response Fig. 7).

Response Fig. 7. Updated main text Fig. 2 showing ADMIXTURE labeling for the central tanglegram.

It is really difficult to figure out where the tanglegram is – I’m used to seeing these at the tips and given the small size of the figure, the dashed lines are hard to see. I wonder if a small inset could be shown to highlight these? Or are the ADMIXTURE crossed bars showing this? If so, an explanation in the legend would be helpful.

We appreciate the difficulty in making out the incongruences, the visuals of which are made challenging by very shallow branchings among the Hawaiian mints. The trees were run through various methods of detangling to find the lowest, least tangled score, but because branches can be rotated, there were in fact many ways to display this. We have now highlighted the dotted lines that represent discordances between the SNP and BUSCO trees using red-colored and thicker lines, and hope this helps readers to identify discordances. The crossed bars in the ADMIXTURE plot indeed do indicate discordance, but not all discordances were possible to show this way. Please refer to updated main text Fig. 2, shown as Response Fig. 7 above.

What do the numbers in parentheses mean after some taxa?

The numbers after the taxon names in Fig. 2 indicate the project ID number for taxa that have more than one individual represented (see Suppl. Table 4) and this information has been added to the caption.

Figure 3 – what data are shown here? “this dataset”

Thank you; we have clarified this statement as follows:

Lines 1186-1187: “... Schematic tree showing hypothesized allopolyploid events among taxa sampled (the circles represent polyploid events).”

Figure 5 b – helpful to indicate in the legend that the numbers along the side refer to the map and transect in a.

Thank you; we have clarified the caption as follows:

Lines 1210-1211: “**b** ADMIXTURE results (dataset HM1), with best-fitting $K = 2$, is shown to the left (numbers refer to the sampling sites shown in **a**)...”

Reviewers' Comments:

Reviewer #1:

Remarks to the Author:

I am fully satisfied with the author's responses. However, all the figures are of very poor resolution, both in the main figure and the supplementary figure. This needs to be updated prior to final acceptance.

Reviewer #2:

Remarks to the Author:

The authors have improved the manuscript and provided clarity on previous points of confusion. I do not have any additional comments.

RESPONSE TO REVIEWERS' COMMENTS

Reviewer #1 (Remarks to the Author):

I am fully satisfied with the author's responses. However, all the figures are of very poor resolution, both in the main figure and the supplementary figure. This needs to be updated prior to final acceptance.

We appreciate the reviewer's comments. We have gone through all figures carefully, including increasing resolution of all figures in the main text that we think are now much improved. We increased the resolution of most supplementary figures to ensure higher resolution images and readability of all figures, but we note that we had to stay within a manageable file size and the maximum limits of 30MB.

Reviewer #2 (Remarks to the Author):

The authors have improved the manuscript and provided clarity on previous points of confusion. I do not have any additional comments.

We thank the reviewer for the positive comments.